# CRAC and SK Channels: Their Molecular Mechanisms Associated with Cancer Cell Development

**DOI:** 10.3390/cancers15010101

**Published:** 2022-12-23

**Authors:** Adéla Tiffner, Valentina Hopl, Isabella Derler

**Affiliations:** Institute of Biophysics, JKU Life Science Center, Johannes Kepler University Linz, A-4020 Linz, Austria

**Keywords:** cancer, CRAC channel, SK3 channel, cancer signaling pathways, cancer hallmarks

## Abstract

**Simple Summary:**

Cell fate is ultimately determined by the precisely coordinated action of the Ca^2+^-signaling machinery. During carcinogenesis, Ca^2+^ signaling is significantly remodeled due to mutations and/or ectopic expression. Here, we summarize current knowledge on how alterations in Ca^2+^ signaling contribute to the development of different cancer hallmarks. Emphasis is placed on the structure/function relationship of the well-studied store-operated Ca^2+^ channel, i.e., Orai1, and the Ca^2+^-activated K^+^ channel, i.e., SK3, alongside their individual and joint roles in cancer. This review lays out the current state of knowledge of Ca^2+^-signaling effectors and proteins as potential targets for the treatment of certain cancer types, with Orai1 and SK3 presented as emerging therapeutic targets.

**Abstract:**

Cancer represents a major health burden worldwide. Several molecular targets have been discovered alongside treatments with positive clinical outcomes. However, the reoccurrence of cancer due to therapy resistance remains the primary cause of mortality. Endeavors in pinpointing new markers as molecular targets in cancer therapy are highly desired. The significance of the co-regulation of Ca^2+^-permeating and Ca^2+^-regulated ion channels in cancer cell development, proliferation, and migration make them promising molecular targets in cancer therapy. In particular, the co-regulation of the Orai1 and SK3 channels has been well-studied in breast and colon cancer cells, where it finally leads to an invasion-metastasis cascade. Nevertheless, many questions remain unanswered, such as which key molecular components determine and regulate their interplay. To provide a solid foundation for a better understanding of this ion channel co-regulation in cancer, we first shed light on the physiological role of Ca^2+^ and how this ion is linked to carcinogenesis. Then, we highlight the structure/function relationship of Orai1 and SK3, both individually and in concert, their role in the development of different types of cancer, and aspects that are not yet known in this context.

## 1. Introduction

Every year, 18.1 million cases of cancer are diagnosed worldwide, of which 9.5 million lead to death. These numbers are expected to rise 1.6-fold by 2040 [1]. The reasons for enhanced cancer-related deaths originate from the complexity of this disease. Generally, human cells grow and divide to form new cells, while the old cells die and are replaced by new ones. Sometimes, however, abnormal or damaged cells continue to grow and proliferate when they should not. The latter can lead to the formation of non-cancerous (benign) or cancerous tumors. Benign tumors can be removed and usually do not grow back, whereas cancerous tumors spread or invade nearby tissues and migrate to other sites in the human body to form new tumors, which is known as metastasis. Despite the complexity of the disease, cancer development can be described by a list of cancer hallmarks defined by Hanahan and Weinberg [2]. The most general hallmarks are sustained proliferation, apoptosis resistance, evading growth suppressors, angiogenesis induction, replicative immortality, tissue invasion, and metastasis [2,3,4,5,6,7]. One essential factor that contributes to cancer progression is calcium (Ca^2+^). In this review, we first describe the general role of Ca^2+^ in tumor development. Since the co-regulation of the Ca^2+^ channel: Ca^2+^ release-activated Ca^2+^ (CRAC) channel, and the Ca^2+^-regulated channel: Ca^2+^-activated K^+^ channel, or SK3, have been extensively reported to play a role in certain cancer types in the last century, we delineate here the current knowledge of the molecular mechanisms of both channels individually and in co-regulation. We highlight critical factors that determine their structure/function relationship, their roles in carcinogenesis identified to date, and outstanding questions in this context.

### 1.1. Physiological Role of Ca^2+^

Ca^2+^ ions are versatile intracellular signals that regulate a plethora of cellular processes including gene transcription, proliferation, and cell migration [8,9]. They act as second messengers linking external or intraluminal signals (endoplasmic reticulum (ER), mitochondria) and lead to intracellular responses through a variety of distinct cascades. Under physiological conditions, the Ca^2+^ concentration on the extracellular side and in intracellular organelles is in the range of mM, and it is 10,000-fold lower in the cytosol [10,11,12,13]. This Ca^2+^ gradient defines the versatility of this signal ion in the life cycle of a cell, to initiate and drive processes such as immune cell activity, neurotransmitter release, or muscle contraction [8,9,14]. Activation of the cell can lead to elevations in intracellular Ca^2+^ levels that occur either through Ca^2+^ release from intracellular stores (ER, mitochondria) or Ca^2+^ influx across the plasma membrane (PM) from the extracellular space [15]. 

Cellular Ca^2+^ signaling is orchestrated by Ca^2+^-transporting and Ca^2+^-sensing proteins. Ca^2+^ level enhancements can be initiated by the stimulation of a membrane receptor (e.g., G-protein coupled receptor (GPCR)), membrane depolarization (voltage-dependent channels [16]), or mechanical stress (mechanosensitive Ca^2+^ channels [17]). Ca^2+^ signaling mediated by membrane receptor stimulation triggers the development of cellular factors (e.g., inositol-tri-phosphate (IP_3_), diacylglycerol (DAG)) that activate Ca^2+^ ion channels in intracellular compartments (e.g., IP_3_R in ER membrane) and/or the plasma membrane (e.g., receptor- or store-operated channels) to allow Ca^2+^ flux into the cytosol. Cytosolic Ca^2+^ elevations are sensed by a number of proteins such as Calmodulin (CaM) and downstream targets including CaM kinase (CaMK), calcineurin (CN), or protein kinase C (PKC) (Figure 1A), which drive various downstream processes, including gene transcription, proliferation, cell death, migration, and metabolism [14]. Ca^2+^ signaling events are terminated by Ca^2+^ transporters that pump Ca^2+^ back into cellular organelles (e.g., SERCA) or to the extracellular side (PMCA) [18]. 

### 1.2. Ca^2+^ Signal Transduction in Major Cancer Hallmarks

In cancer cells, genetic and epigenetic alterations can lead to the remodeling of Ca^2+^-signaling components, disrupting the healthy Ca^2+^ balance [3,6,19,20]. This allows cells to bypass mechanisms controlling inappropriate proliferation and prevent the survival of ectopically proliferating cells outside their normal niches [21]. Ca^2+^ acts at different stages of cancer signaling cascades either directly via Ca^2+^ signaling proteins (CaM, CaMK, CN) or indirectly via transcription factors (e.g., NFAT) or oncogenic routes. The two major oncogenic pathways [22] that have a central role in the development of the different cancer hallmarks are the rat sarcoma virus—extracellular-signal-regulated kinase (Ras-ERK)—and phosphoinositide 3-kinase (PI3K)-Akt (Akt, also known as protein kinase B, PKB) pathways. They are activated upon ligand binding to integrin adhesion receptors and signaling by cytokines, hormones, or exogenous growth factors [23]. Within the Ras-ERK pathway, stimulation of the growth factor receptor tyrosine kinase (RTK; e.g., epidermal growth factor receptor (EGFR)) activates the small GTPase, Ras, followed by the serine/threonine kinase (Raf), and finally the extracellular signal-regulated kinase, ERK [21]. In the PI3K-Akt pathway, receptor stimulation triggers the activation of the lipid kinase (PI3K), which in consequence activates the serine/threonine kinase Akt. Both ERK and Akt phosphorylate various downstream effectors, including transcription factors and kinases [24]. (Figure 1A,B). Ca^2+^ and these oncogenic pathways can affect each other in a bidirectional manner to promote cancer progression. On the one hand, distinct Ca^2+^ signals can tune oncogene-dependent signaling [18,25]. For example, the activation status of the Ras oncogene is altered by an interplay of Ca^2+^ ions with Ras regulatory factors, such as CaM [25,26]. On the other hand, oncogene-regulated routes can reshape Ca^2+^ signals, as evidenced by the fact that the oncogene Ras interacts with and activates PLCε to produce IP_3_, which triggers ER-Ca^2+^ store-depletion [23] (Figure 1). Overall, direct and indirect modulation of cancer signaling pathways promotes cancer cell proliferation, survival, and migration, as described in detail in the following subsections.

### 1.3. Ca^2+^-Dependent Dysregulation of Proliferation

The multitude of Ca^2+^-dependent effectors modulating proliferation is selectively and efficiently controlled by specific spatiotemporal Ca^2+^ signaling. The abovementioned ways of Ca^2+^-dependent regulation of cancer signaling pathways control proliferation at the level of the cell cycle machinery. Particularly, Ca^2+^ signaling at the onset of the G1 phase leads to the activation and expression of transcription factors of Nuclear Factor Activated T cells (NFAT), cAMP-responsive element binding protein (CREB), and AP1 (FOS, JUN) families [25]. These components coordinate the expression of cell cycle regulators, notably certain types of cyclin proteins (Cyclin D/E) and cyclin-dependent kinases (CDK2/4/6). Ca^2+^ also drives their complex formation (CDK4/6-Cyclin D, CDK2-Cyclin E) at the end of G1 to finally ensure the transition to the S phase. The progression of the G1 phase is further fine-tuned by CDK inhibitor proteins (p21, p27), whose action is controlled directly by Ca^2+^ via Ca^2+^-sensing proteins, CaM, CaMKII, and calcineurin (CN) and the tumor suppressor p53 [25]. Ca^2+^ is also essential for subsequent cell cycle phase transitions (G1/S, G2/M) and associated rearrangements of centrosomes are triggered by Ca^2+^ oscillations acting in concert with CaM and CaMKII [18,27] (Figure 1A).

Further upstream, Ca^2+^ impacts the cell cycles via Ras-ERK or PI3K-Akt cascades. For instance, Ca^2+^ induces ERK phosphorylation via the CaM-CaMKII pathway to regulate proliferation [28]. This occurs via ERK-mediated phosphorylation of various transcription factors essential for proliferation, most notably Myc. Myc triggers the expression of a number of proteins (e.g., cyclins, CDKs) that interfere with the cell cycle [27]. In addition, Ca^2+^ might also intervene in the complex action of Akt, which governs proliferation during cell cycle progression, by promoting protein synthesis essential for cell growth, suppressing cell cycle inhibitors through their sequestration or impaired gene transcription, and controlling a set of enzymes involved in the G2/M transition [29] (Figure 1A).

A variety of other cancer signaling pathways (e.g., Wnt/ß-catenin, Wnt/Ca^2+^) target Myc to modulate cell cycle progression [21,30], among which the Wnt/Ca^2+^ route can control Ca^2+^ signaling. The Wnt-Ca^2+^ pathway is initialized by Frizzled receptors, which initiate a classical G-protein-coupled signaling cascade that results in the production of IP_3_ and DAG and is thus directly linked to Ca^2+^ signaling pathways [30,31].

### 1.4. Ca^2+^-Dependent Dysregulation of Cell Survival and Cell Death

Cancer cells harness the Ca^2+^ signaling machinery to ensure their survival and protect themselves from apoptosis. Ca^2+^ is involved in the activation of pro-survival signaling pathways and anti-apoptotic proteins that inhibit or neutralize death signals [25]. In this context, the ER and mitochondria are the major locations to determine cell fate. Although elevations in cytosolic Ca^2+^ due to ER store depletion are essential for many vital processes, they can also trigger apoptosis. For example, cytosolic death effectors such as calpain are stimulated, rendering the cell susceptible to apoptosis through the activation of caspases [32]. Moreover, Ca^2+^-flux between the ER and mitochondria can stimulate death-inducing signals [18,25]. Several tumor suppressors, such as p53, are enriched at the ER. There, p53 interacts with SERCA pumps, increases the ER’s Ca^2+^ load, and contributes to Ca^2+^ crosstalk between the ER and mitochondria. This enables the efficient release of pro-apoptotic factors [33,34,35,36,37]. p53 and Ca^2+^ ions act as interdependent cellular signals, although the detailed cellular pathways are unknown [38] (Figure 1B). A recent study [39] has shown that the protein kinase CK2 plays a critical role in maintaining elevated cytosolic Ca^2+^ levels that promote prostate cancer progression. CK2 inhibition reduced cytosolic Ca^2+^ and increased Ca^2+^ levels in the ER and mitochondria to induce apoptosis [39], whereas the underlying mechanisms remain to be clarified.

Whether apoptosis occurs is further defined by a complex interplay of pro- (e.g., Bim—BCL2—interacting mediator of cell death, BID-BH3-interacting domain death agonist, Bad—BCL2-associated agonist of cell death) and anti-apoptotic (e.g., BCL2, BCLxL) factors at the mitochondria [25,32]. Dysregulation of Ca^2+^ signaling routes, either directly via Ca^2+^ sensing proteins or indirectly via oncogenic signaling pathways, can lead to an imbalance between pro- and anti-apoptotic regulators in favor of anti-apoptotic proteins [11,18,25,32,40,41,42,43] (Figure 1B). Most cancer cells display enhanced expression levels of apoptosis-regulating proteins of the BCL2 family. They are responsible for reducing the amount of Ca^2+^ in the ER or preventing Ca^2+^ uptake into mitochondria, which can lead to apoptosis resistance. Apoptosis can be also reduced by the interplay of Akt with IP_3_R, leading to decreased Ca^2+^ release from the ER [18,25] (Figure 1B).

### 1.5. Ca^2+^-Dependent Dysregulation of Migration and Invasion

Cancer metastasis involves epithelial–mesenchymal transition (EMT), cell migration, invasion, angiogenesis, and intravasation, all of which are controlled by Ca^2+^ [18,25]. EMT is a gradual, transient process (already important in embryonic development) in which cell–cell connections degrade and detach from the basement membrane, causing cells to lose polarity [44]. In several cancer cell types, it is associated with the downregulation of common epithelial genes (cytokeratins, E-cadherin) and the upregulation of mesenchymal markers (vimentin, fibronectin, N-cadherin, and metastasis-associated in colon cancer-1 (MACC-1)) [42,44,45,46,47,48,49,50,51,52,53] (Figure 1C, Top). This switch in markers during EMT is linked to aberrant Ca^2+^ signaling in many cancer types [19,32]. For instance, CaMKII activates vimentin via phosphorylation [54]. Furthermore, abnormal Ca^2+^ levels develop due to the GPCR signaling and cancer-specific upregulation of certain types of Ca^2+^ ion channels. Consequently, elevated Ca^2+^ levels activate the transcriptional machinery, vital for the upregulation of certain genes such as Snail, Twist, Zeb, and N-cadherin. These genes participate in numerous transitions, including cell polarity, cytoskeletal remodeling, migration, and invasion [40,42,51,54,55]. Moreover, genes activated after signal transduction through the Wnt/Ca^2+^ pathway govern cell migration by targeting ß-catenin [30] (Figure 1C).

Directional movement of migratory cells is usually driven by highly localized Ca^2+^ signals reaching a rear-to-front Ca^2+^ gradient. Rear-end retraction of the migratory cell is driven by Ca^2+^-modulated myosin light chain kinase (MLCK), followed by actomyosin contraction and disassembly of focal adhesions. The latter is governed by the Ca^2+^-sensitive protease calpain, which cleaves focal adhesion kinase (FAK), vinculin, talin, and integrins to disconnect the membrane from the cytoskeleton. Conversely, directed actin polymerization and growing focal adhesions located at the leading edge of migrating cells to form membrane extension are stimulated by spatially confined Ca^2+^ signals. This is modulated by small GTPase, such as Ras and Rac1, Ca^2+^-dependent factors, including CaMK and proline-rich tyrosine kinase 2 (PYK2), and FAK [32] (Figure 1C). Altered Ca^2+^ levels in tumor cells can further promote FAK disassembly and contribute to the reduction of cell adhesions that promote cell migration. Moreover, the formation of membrane extensions (invadopodia) is driven by the expression of Ca^2+^-dependent pro-invasive enzymes (e.g., matrix metalloproteins (MMPs)) that contribute to the degradation of the extracellular matrix (ECM) [18,32,40] (Figure 1C).

### 1.6. Ca^2+^-Dependent Dysregulation of Other Cancer Hallmarks

Another phenomenon that occurs in most malignant tumors is non-physiological oxygen levels, known as hypoxia. Cell adaptation to hypoxia is primarily regulated by overexpression of the transcription factor HIF-1, which leads to angiogenesis, cell proliferation/survival, and invasion/metastasis [56,57,58]. Hypoxia can increase transcription and expression of Ca^2+^ channels and is therefore frequently connected to cytosolic Ca^2+^ enhancements, which can promote the transcription of target genes responsible for the development of various cancer hallmarks [59]. Indeed, chemical treatments leading to hypoxia of different cancer cell lines increased intracellular Ca^2+^ as well as the expression of certain Ca^2+^ ion channels [60,61]. Moreover, silencing of the HIF-1 transcription factor in breast cancer reduced the expression of critical Ca^2+^ ion channels [62]. Notch is a cellular pathway associated with HIF-1 signaling. Knock-down or pharmacological inhibition of Notch-1 decreased the expression of certain Ca^2+^ ion channels in breast, colon, and glioma cancer cells [63,64]. Conversely, Ca^2+^ fluxes from the ER and outside of the cell can control HIF-1 signaling [24,59] (Figure 2). However, the underlying mechanisms of these effects are only starting to be understood. Overall, hypoxia is responsible for the development of different cancer hallmarks, alters cancer cell metabolism, and contributes to therapy resistance [56].

## 2. Ca^2+^ Ion Channels and Their Role in Cancer

Intracellular Ca^2+^ is integral in the pathogenesis of key cancer features, as described in the previous subsection. Considering that many current anticancer drugs are largely ineffective in many cancers, emerging scientific discoveries indicate that the pool of Ca^2+^-permeable and -dependent ion channels are a rich supply of new potential therapeutic targets. Already in the late 1980s, a distinct pattern of functional expression of ion channels in cancer cells was detected, providing a possible link between ion channels and carcinogenesis [65,66,67,68,69,70]. Disrupted expression and/or dysfunction of ion channels can deregulate cellular processes that develop into cancer hallmarks [2,4,5,6,7].

Among the diversity of ion channels, PM ion channels have a significant role in the development of various cancer phenotypes and can be categorized as voltage-gated (VGCC) and non-voltage Ca^2+^ ion channels [32,40]. Despite their main role in ‘‘excitable cells’’, VGCC are proposed to govern some common molecular mechanisms of carcinogenic events due to their overexpression in many cancers [71,72]. Non-voltage-gated channels can be subdivided into ligand-gated channels (LGC, e.g., purine ionotropic P2X receptor families) [73], receptor- or second messenger-operated channels (ROC or SMOC, e.g., some transient receptor potential (TRP) channel members and the Orai family) [74,75], SOC (some TRP channel members and the Orai family) [74,76], acid-sensing channels (ASIC) [77], and mechanically gated channels [78]. A number of studies reported their altered expression and/or function in different tumor cells, driving the development of cancer hallmarks [53,79,80,81,82,83,84,85,86,87,88,89]. Particularly, TRP channels perform versatile functions in tumorigenesis, in addition to their multiple roles in the healthy body [90,91,92]. They sense changes in the local environment and can be activated by a set of physical and chemical stimuli, making some of them key elements of tumorigenesis. For example, TRPC6 and TRPV6 govern NFAT translocation that is critical for proliferation. Mechanosensitive TRPV2 and TRPM7 determine cancer cell migration and invasion [40].

Due to the essential role of the most studied store-operated Ca^2+^ ion channel, the CRAC channel, in the co-regulation with Ca^2+^-dependent K^+^ ion channels in cancer, we focus initially on its currently known structure/function relationship as well as its individual role in cancer. It is composed of the stromal interaction molecule, STIM, containing its Ca^2+^ sensor embedded in the ER lumen [93,94,95,96], and the highly Ca^2+^-selective ion channel Orai in the PM [97,98,99,100,101,102]. Emerging evidence reveals that STIM and Orai proteins are predominant Ca^2+^ entry mechanisms in most cancer cells [80,103,104,105,106,107,108,109,110] and promote various cancer hallmarks [80,103,104,105,106,108,109,111].

### 2.1. CRAC Channels

CRAC channels, composed of STIM and Orai proteins, are distinguished by their activation mechanism via the release of Ca^2+^ from the ER. Precisely, ligand binding to PM receptors leads to the activation of G-proteins, which initiate the production of phospholipase C to generate IP_3_ from phosphatidylinositol-biphosphate (PIP_2_). Subsequently, IP_3_ couples to the IP_3_ receptor in the ER membrane, which triggers ER-Ca^2+^ store depletion [74,112,113,114,115,116,117]. The latter induces the conformational change and oligomerization of STIM proteins [110,118] and consequently their coupling to Orai Ca^2+^ ion channels [119] in the PM [120,121] (Figure 3A). In a healthy body, STIM and Orai proteins are mainly involved in immune cell function [97], but they also contribute to the regulation of muscle cells or brain function [80,103,104,105,106,108,109,111,122,123,124,125,126,127].

#### 2.1.1. STIM Proteins

The STIM protein family, including the two homologs STIM1 and STIM2 (also called STIM2.2) [118,124,128] (Figure 3B), is further enriched by the corresponding splice variants STIM1L, STIM1A, and STIM1B and STIM2.1 and STIM2.3, respectively [129,130,131,132]. Of the STIM protein family, STIM1 and STIM2 in particular are ubiquitously expressed in many tissues [111,124,128,129,130,131,133,134,135]. In the following, we will focus on STIM1 and STIM2, because only these two isoforms have been described as playing a role in cancer.

In general, the STIM structure consists of a single-pass TM domain with the N-terminus embedded in the ER lumen and its long C-terminal tail exposed to the cytosol. The N-terminus senses fluctuations in Ca^2+^ levels, and the C-terminus relays the activation signal to Orai1 in the PM [119]. The STIM1 N- and C-terminus and the TM domain contribute to structural rearrangements upon STIM’s activation [136,137,138] (Figure 3B).

Specifically, the luminal side of STIM1 comprises the Ca^2+^-sensing EF-hand domain in a complex with the sterile alpha motif (SAM) region. In the resting state, the EF-SAM domain is folded into a compact structure that is stabilized by hydrophobic interactions [139]. Upon Ca^2+^ store depletion, Ca^2+^ ions dissociate, leading to destabilization of the EF-SAM complex, which initiates signal transmission to the C-terminus [140,141,142,143,144].

Next, the activation signal of Ca^2+^ store depletion is conveyed from the N- to the C-terminus via the STIM1 TM domain. Two TM domains within a STIM1 dimer are considered to form a crossing angle that alters upon activation [142]. In support, a cysteine crosslinking screen uncovered that in the resting state, only the C-terminal portions of the STIM1 TM domains are in close proximity. In the active state, the N-terminal TM segments are closer together, possibly changing the crossing angle of the cytosolic C-termini [140].

Upon Ca^2+^ store depletion, the STIM1 C-terminal domain undergoes an extensive conformational rearrangement to change from a folded quiescent to an extended state, which leads to the exposure of STIM1 oligomerization and Orai1 coupling sites. The STIM1 C-terminus is composed of three typical protein–protein interaction domains known as coiled-coil regions (CC1, CC2, CC3), the inhibitory [145] or CRAC modulatory domain [146], the microtubule end-binding domain (EB), the Ser/Pro-rich region [94], and the lysine-rich region [147,148] (Figure 3B).

In the closed state, STIM1 is locked by intramolecular interactions between the C-terminals CC1 and CC3. The inhibitory clamp formed by a segment of CC1 (CC1α1) and CC3 of monomers within a dimer is further modulated by intrahelical interactions within two other parts of CC1 (CC1α2 with CC1α3) [149,150,151]. Upon activation, this intramolecular inhibitory clamp is released, and the STIM1 C-terminus changes into an extended conformation, which is stabilized by homomeric interactions between CC1 and CC3 of different STIM1 proteins [100,142,152,153]. Thereby, the STIM1-Orai1 coupling site is released.

A minimal portion of the STIM1 C-terminus called CAD (CRAC-activating domain) or SOAR (STIM1-Orai1-activating region) is sufficient for coupling to and activation of Orai channels. They include mainly CC2 and CC3 [99,102,154,155]. Currently, the structural resolutions of two slightly distinct SOAR-like fragments are available [149,152,156,157]. Both structures indicate that two SOAR monomers are arranged in an anti-parallel manner to form a dimer. Despite the essential inter- and intramolecular interaction sites being uncovered, the overall conformation of the two dimeric structures is relatively distinct [156,157]. A recent single-molecule FRET approach revealed more similarities with the CC2-CC3 crystal structure than the CC1α3-CC2 NMR structure [150], but further structural studies of STIM1 fragments or full-length STIM1 are highly awaited. In the quiescent state, the inhibitory clamp hides the CAD/SOAR region for coupling to Orai1. Critical Orai1 coupling sites are supposed to be located at the connection of CC2 and CC3, the so-called apex (F394) [158], and the N-terminal (L373, A376) [159] (Figure 3B). In the tightly packed, inactive conformation, the apex is oriented to the ER [150,158]. STIM1 activation is assumed to lead to an unfolding of the C-terminus, thus allowing the apex to reach the PM, couple to Orai1, and finally activate Ca^2+^ influx.

#### 2.1.2. Orai

Orai proteins activated by STIM1 function as highly Ca^2+^-selective ion pores in the PM. The Orai protein family comprises three human Orai paralogs, namely, Orai1–3 (Figure 3C) [121,160]. All three isoforms are ubiquitously expressed in many tissues [135,161,162,163]. RNA transcripts of Orai1 and Orai2 are found primarily in the spleen, lymph nodes, appendix, bone marrow, and brain, whereas Orai3 is detected more in the prostate, placenta, ovaries, testis, adrenal, urinary bladder, thyroid, endometrium, kidney, liver, and many other tissues [135,161,162,163]. Of note, the expression of Orai3 is restricted, because it is found only in mammals [164].

Orai proteins consist of four TM domains (TM1–TM4) connected by two extracellular and one intracellular loop and flanked by a cytosolic N- and C-terminus [165,166,167]. The N-, C-terminus and the intracellular loop region (loop2) [168,169] are crucial for STIM1-mediated Orai1 activation, whereas the Orai1 C-terminus (L273) is the main STIM1 coupling site [101,138,170,171,172]. The STIM1/Orai1 association pocket (SOAP) has been resolved by NMR with fragments of STIM1 (aa 312–387) and Orai1 (aa 272–292) [156]. The identified key sites responsible for STIM1/Orai1 coupling are in line with experimental findings [154,156,173]. Additionally, the loop2 region functions as a STIM1-Orai1 gating interface after the functional coupling of STIM1 to the Orai1 C-terminus [99,136,174]. As for the N-terminus, functional results indicate that it contributes to STIM1-dependent Orai1 activation; however, it is still a matter of debate whether it forms a direct STIM1 coupling site [10,100,136,137,138,156,159,169,175,176].

Structural resolutions of the *Drosophila melanogaster* Orai (dOrai) and corresponding gain-of-function (GoF) dOrai (dOrai H206A, dOrai P288L) mutants consistently exhibit a hexameric assembly of Orai subunits. Based on the high homology of dOrai to human Orai1 (hOrai1) within the TMs, it is assumed that hOrai1 also forms a hexamer. Of the four TM domains, the first TMs (TM1s) form an inner ring lining the pore in the center of the channel [119,177,178,179]. This is composed of a Ca^2+^ accumulation region at the extracellular side followed by the selectivity filter (E106), the hydrophobic cavity (F99, V102), and at the cytosolic side, the basic region. Interestingly, TM1 expands helically approximately 20 Å into the cytosol, forming an extension of the Ca^2+^ ion pore [10,112,113,114,119,136,180]. This cytosolic region constitutes the last third of the Orai N-terminus, which is also referred to as the extended TM Orai N-terminal (ETON; aa 73–90) region (Figure 3C) [136]. The pore-lining helix is surrounded by TM2s and TM3s forming a middle ring and by TM4s as the outer ring of the channel complex [119]. In addition to the Ca^2+^ ion pore, another essential feature of the Orai1 complex represents its periphery, composed of TM4 and the C-terminus [138,170]. The closed-state structural resolutions exhibit a kink at P245 in TM4 and a bent connection to the helical C-terminus [119,170]. The open state structures suggest conformational rearrangements in these areas, although the extent of these structural changes is still a matter of debate. Nevertheless, this information led to the hypothesis that STIM1 coupling to its main coupling site within Orai1, the C-terminus, and its subsequent interplay with the loop2 allosterically connected to the nexus region [169,170,174], triggers pore opening via a wave of interdependent TM domain motions [181,182]. The latter is supported by the fact that a set of positions within all TM domains (e.g., H134, V181, P245) function as gating checkpoints, as their mutation can lead to either gain- (GoF) or loss-of-function (LoF), depending on the amino acid substitution. Moreover, several LoF mutations act dominantly over GoF mutations in various combinations, proving that a global conformational change of the channel complex is essential for pore opening [181,182].

The general structure of the three Orai isoforms is comparable, yet they have an overall sequence identity of 50–60%, with TM1 being identical among the three Orai proteins, whereas the other TM domains are approximately 81–87% similar (Figure 3C). The cytosolic and extracellular regions exhibit greater differences, with the N-terminus (aa 1-90) showing 34% and the C-terminus (aa 265–301) 46% sequence identity [164]. For the extracellular (loop1, loop3) and intracellular (loop2) loop regions, loop1 is 60–80% [147], loop2 is 80–90% [169], and loop3 is only 20–30% conserved [164]. These differences account for a number of isoform-specific functional differences, such as current size, inactivation, or binding affinity to STIM1 [10,114,136,169,183,184,185,186] and might be promising targets for potential pharmacological interference [170,171].

In summary, CRAC channel activation represents a multistep activation cascade involving STIM1 unfolding, STIM1 oligomerization, STIM1-Orai1 coupling, and Orai1 activation. Resolving these different intermediate activation states could help researchers to find new targets for selective therapeutic strategies also in cancer. In particular, the isoform-specific differences of Orai channels might be promising in cancer-type-specific drug discovery and therapy development [185]. Nevertheless, several aspects remain still to be clarified, such as whether STIM1 also binds to the N-terminus of Orai1, the detailed STIM1/Orai1 binding pockets aside from the already known one formed by their C-termini, the stoichiometry of the STIM1/Orai1 complex for maximal activation, and additional isoform-specific differences in the structure/function relationship.

#### 2.1.3. CRAC Channels and Cholesterol-Rich Regions

Increasing evidence reveals that membrane proteins are localized in microdomains containing especially cholesterol, sphingolipids such as sphingomyelin, and glycosphingolipids [187]. These cholesterol-rich regions provide platforms required for membrane protein sorting and the assembly of signaling machinery, thus dictating protein–lipid and protein–protein interactions. In cancer, too, the remodeling of Ca^2+^ ion channels may involve their altered arrangement and interplay in the membrane due to structural rearrangements of the channel, assembly of the channel complex, or channel interaction with regulatory proteins that affect their function [3]. In the context of lipid-driven regulation of STIM1 and Orai1, there is increasing evidence that the function of the STIM1/Orai1 complex is modulated by/within cholesterol-rich regions [188,189], although STIM1 and Orai1 are sufficient to form the CRAC channel [190]. This dependence on lipids is not surprising, given the coincidence of STIM1-Orai1 coupling and their activation at ER–PM junctions [191]. The function of STIM1 and Orai1 is affected by direct interaction with some lipids, including phospholipids PIP_2_ and PI4P, cholesterol, and sphingomyelin [192,193,194,195,196,197].

Specifically, PIP_2_ modulates STIM1 function via direct binding to its C-terminal end, a lysine-rich region [99,198,199,200,201]. This allows STIM1 to stably interact with PIP_2_ and PIP_3_ located in cholesterol-rich regions in the PM [189,199,202,203,204,205]. Hence, STIM1 first couples with PM-localized PIP_2_ and PIP_3_ in cholesterol-rich regions before interacting directly with Orai1 [206]. The N-terminus of Orai1 also contains a polybasic region that is sensitive to PIP_2_ and is therefore essential for modulating Orai1 in membrane domains with different PIP_2_ content [207]. However, it is currently difficult to analyze the modulatory role of PIP_2_ on Orai1 due to its dependence on STIM1. Moreover, the precursor of PIP_2_, PI4P, regulates CRAC channel function [203,208,209], but the detailed mechanisms remain to be determined.

Cholesterol affects CRAC channel function via direct interaction with STIM1 and Orai1. Both CRAC channel proteins contain a cholesterol-binding motif, namely, the cholesterol recognition amino acid consensus motif (-L/V-(X)(1-5)-Y-(X)(1-5)-R/K-; X represents from one to five any amino acids before the next conserved residue) [210,211,212,213]. In STIM1, it is located in the C-terminus (aa 357–366) [195], and in Orai1, it is formed by aa 74–83 in the ETON region [113]. In both cases, the mutation of key residues therein, which have been demonstrated to disrupt cholesterol binding, led to enhanced store-operated STIM1-mediated Orai1 currents, in accordance with the effect observed upon cholesterol depletion. Overall, this suggests that STIM1 and Orai1 coordinate analogous cholesterol-dependent mechanisms of CRAC channel regulation. Regarding the impact of chemical cholesterol depletion, there are still conflicting results. Interestingly, though cholesterol depletion by methyl-β-cyclodextrin (MßCD) reduced endogenous SOCE [214], cholesterol oxidase- or filipin-induced reduction in membrane cholesterol enhanced endogenous SOCE [113]. Moreover, MßCD internalizes Orai1, which could be rescued by caveolin (Cav-1) overexpression, a key component of cholesterol-rich regions [196]. Interestingly, other studies have reported that MßCD application to cells overexpressing STIM1 and Orai1 had either no inhibitory [193] or enhancing effects on STIM1-mediated Orai1 currents [195]. Such distinct effects could potentially occur due to distinct expressions of the respective proteins. Alternatively, these observed differences are probably attributable to the milder manipulation of cholesterol levels or distinct membrane composition due to the application of cholesterol oxidase or filipin compared to MßCD.

Sphingomyelin, which is also abundant in cholesterol-rich regions, controls CRAC channel function. Indeed, the application of sphingomyelinase (SMase) D diminished CRAC channel currents without impacting Ca^2+^ store depletion. However, whether this modulatory role occurs due to direct binding or allosterically is still unknown [215,216].

Additionally, STIM1 and Orai1 undergo S-acylation, the posttranslational tethering of a medium-length fatty acid, to a cysteine—in these cases, C437 in STIM1 and C143 in Orai1. S-acylation of STIM1 controls its puncta formation and maximal activation of CRAC channels [217]. In Orai1, S-acylation is mediated by the protein acyl transferase (PAT) 20 and controls Orai1 trafficking, activation, and maintenance of its accumulation in cholesterol-rich domains essential for downstream signaling [197].

Additionally, perturbations to plasma membrane lipids affect other proteins that impact STIM1 and Orai1 expression and function. These include a variety of lipid- or ER-PM transition-dependent accessory proteins at the ER–PM contact sites [218], as we recently reviewed in detail [187]. Briefly, these include proteins situated in the ER that can establish direct or indirect interactions with the PM, namely, the Extended-synaptotagmins (E-Syts) with E-Syt 1-3 [219], GRAMD2A [220], and Anoctamin 8 (ANO8) [221]. They are involved in controlling the formation of ER–PM contact sites as well as the lipid composition, especially PIP_2_, of the membranes, thereby having various effects on the modulation of STIM1/Orai1 coupling and function. Additionally, ER- or PM-associated proteins located in the ER-PM junctions, in particular, septins [222,223,224,225,226,227], junctate [228,229], RASSF4 [229], STIMATE [230], SARAF [205,231,232], and Cav-1 [196,233,234,235,236] modulate the interplay of STIM1 and Orai1. Furthermore, the interplay of lipids with channels, as reported for TRPC1, affects the function of CRAC channel components. Though Cav-1-dependent translocation of TRPC1 into cholesterol-rich regions allows its store-operated activation via STIM1 binding [237,238,239,240], in the absence of Cav-1, TRPC1 is moved out of cholesterol-rich regions to function as an agonist-dependent and STIM1-independent channel [241,242,243].

The fact that the CRAC channel complex is not a closed entity but can be modulated by a variety of proteins and lipids in the vicinity of the ER–PM contact sites highlights the multiple regulatory possibilities that may also play a role in the pathogenesis of cancer and could be exploited for therapeutic applications.

### 2.2. CRAC Channels and Cancer

Although CRAC channels are one of the most important pathways for the cellular Ca^2+^ influx to maintain healthy body functions [244], there is increasing evidence that they are a major source of Ca^2+^ influx into cancer cells, where they are linked to tumorigenesis. This is due to either altered expression of CRAC channel proteins [81,83,103,104,105,108,109,111,122,126,245,246], specifically, STIM1/STIM2, Orai1, and Orai3 in cancer cells compared with healthy cells or mutations in STIM1 or Orai1 proteins. Such dysregulations can promote the proliferation, migration, invasion, and metastatic spread of cancer cells and be responsible for a poor prognosis and high mortality rate of patients suffering from certain cancer types [85,86,247,248,249,250,251] (Figure 4A–C).

#### 2.2.1. Cancer Associated with Distinct Expression Levels of CRAC Channel Proteins

Distinct expression levels of STIM and Orai compared to corresponding healthy tissues have been detected in cells of the breast [82,252,253], cervical [248], colorectal [85,86,254], esophageal [255], gastric [53], glioblastoma [88,256], hepatocellular carcinoma [251,257], leukemia [87,89], liver [251,258], lung [84,259,260], multiple myeloma [261,262], ovarian [263,264], pancreatic [265,266], and prostate [80,267,268,269,270] cancer and others [83]. Whether CRAC channel proteins are up- or downregulated to promote cancer development and progression depends on the cancer type [80,82,104,249,253,255,259,264,265,266,271]. However, in several tumor cells, upregulation of one or more of the CRAC channel components, mostly STIM1 and Orai1, promotes the development of cancer hallmarks [80,82,104,249,252,253,255,259,264,265,266,271,272] (Table 1).

Notably, other molecular components also contribute to STIM- or Orai-dependent cancer progression. Altered expression and function of the molecular components of SOCE include not only CRAC channel components, such as Orai1 and Orai3, but also other Ca^2+^ ion channels, for instance, the canonical transient receptor potential channels, which have been found in breast cancer (TRPC1 and TRPC6) [104] and ovarian cancer (TRPC1, TRPC3, TRPC4, TRPC6) [263]. Chronic lymphocyte leukemia (CLL) is associated with dysregulated Ca^2+^ signaling in dependence with not only STIM1 and Orai1, but also TRPC1 [89]. Moreover, in the human hepatoma cell line (Huh-7), Ca^2+^ entry occurs via a molecular complex of TRPC6, STIM1, and Orai1 controlling cell proliferation [258]. In the breast cancer cell line, MCF-7, a novel STIM1-independent mechanism for triggering Ca^2+^ entry through Orai1 activation via the accessory protein Secretory Pathway Ca^2+^-ATPase 2 (SPCA2) has been identified [278]. Physiologically, SPCA2 is known to function as a Golgi Ca^2+^ pump, where it is involved in protein glycosylation, sorting, and processing. However, in this cancer cell line, there is a direct interaction of the N- and C-termini of SPCA2 with Orai1 at the cell surface, which controls EMT [83,278]. These cancer cell-specific coregulations of ion channels offer the possibility of finding selective, cancer-type-specific therapeutic targets. In the following, the current knowledge on the role of STIM and Orai proteins in tumor development is summarized.

#### 2.2.2. Proliferation

In several tumor types, enhanced proliferation correlates with ectopic STIM1/Orai1 expression [40,53,79,81,88,249,252,256,258,263,266,276,277]. In some cases, either STIM1, Orai1, or both have been shown to impact certain phases in the cell cycle. Supportively, downregulation of STIM1 and/or Orai1 results in cell cycle arrest at either G0/G1 (glioblastoma [256]), G1/S (cervical [274], gastric [53], cancer) or G2/M transitions (cervical [248], esophageal cancer [255]) due to altered expression of corresponding cyclins, cyclin-dependent kinases, and cell cycle division phosphatases (G1/S: cyclin D1/CDK4 [53,88]; cyclin E/CDK2 [274], cdc25 [248]; G2/M: cyclin B1/cdc2 [255]), and/or proliferation inhibitors ((p. 21 [248]), (p. 27 [255])). Interestingly, in non-small-cell lung cancer (NSCLC) cell line A549, down-regulated Orai1 expression promoted proliferation. Indeed, overexpression of Orai1 led to decreased cyclin D expression and cell cycle arrest in G1 [79,279]. In addition, it abolished another oncogenic pathway modulating proliferation, namely, the EGF proliferative effect along with inhibited Akt phosphorylation [79,279]. In melanoma cells, STIM1- and Orai1-dependent regulation of proliferation occurs through the CaMKII/Raf-1/ERK signaling pathway [276] (Figure 4A). Interestingly, a correlation between STIM1 and Orai1 expression and specific cell cycle regulators has been found in most cancers, providing targeted therapeutic options. However, in several cancer types, it remains unexplored how Ca^2+^ entry via the STIM1/Orai1 machinery affects the various cell cycle effectors, either directly via Ca^2+^-binding proteins or indirectly via transcription factors, Ras/ERK, or PI3K/Akt pathways.

In some cancer types, Orai3 and/or STIM2 expression exerts pro-proliferative effects by influencing the cell cycle via various signaling cascades (Figure 4A). In ER^+^ breast cancer cells, SOCE is mediated exclusively by Orai3 and STIM1/STIM2, whereas in ER^-^ breast cancer cells, the canonical CRAC channel consists of STIM1/Orai1, which is sufficient to trigger SOCE [126]. Specifically, in the estrogen receptor-α (ERα) expressing (ER^+^) breast cancer cells (MCF-7), Ca^2+^ entry via Orai3 targets the proto-oncogenic transcription factor c-myc via the MAPK (originally named ERK) pathway to trigger cell proliferation arrest [126], contributing to ER^+^ breast tumorigenesis [127]. Indeed, Orai3 silencing resulted in decreased c-myc activity and ERK levels and G1 phase cell cycle arrest [126]. In pancreatic ductal adenocarcinoma, Orai3 knock-down resulted in decreased cell proliferation due to a halted cell cycle in the G2/M-phase [266]; however, the underlying mechanism for Orai3-dependent cell cycle regulation is unknown.

In other cancer types, enhanced Orai3 expression increased Orai1/Orai3 heteromeric formation and thus reduced the number of functional Orai1 channels [269] (Figure 4A). In prostate cancer biopsies, AA-mediated activation of Orai1/Orai3 channels enhanced intracellular Ca^2+^ concentration, which controls cell proliferation via Ca^2+^/CN-dependent activation of the transcription factor NFAT. Interestingly, in mouse models, Orai3 silencing controls the G1/S phase cell cycle by decreasing cyclin D1 expression. Conversely, Orai3 overexpression promoted proliferation [269]. Another study [267] suggested heteromeric Orai1/Orai3 channel formation, whereby Orai3 expression is downregulated in cancer, thus altering the Orai1/Orai3 ratio. Higher Orai3 expression, elevated noncanonical Orai1/Orai3 channel formation and controlled proliferation via cell cycle progression of non-small-cell lung adenocarcinoma [249,259]. In support, Orai3 silencing resulted in downregulated cyclin D1 and E expression and reduced Akt phosphorylation, which is associated with reduced proliferation and G1-phase cell cycle arrest [249]. This particular isoform-specific role of Orai3 in different cancers together with isoform-specific features and functionality [185] provides the opportunity for more targeted therapeutic developments.

#### 2.2.3. Cell Survival and Cell Death

Apoptosis resistance is another cancer hallmark connected to up- or downregulated expression and function of CRAC channel components.

The androgen-independent stage of prostate cancer, which represents an aggressive phenotype, is manifested by downregulated Orai1 expression, which leads to apoptosis resistance. In accordance, SOCE is abrogated in these androgen-independent prostate cancer cells [80]. Supportively, overexpression of Orai1 not only restored SOCE but also induced a similar rate of apoptosis in an aggressive type of prostate cancer cells compared to androgen-dependent cells [80]. It is therefore assumed that androgen plays a role in regulating Orai1 expression. Noteworthy, sequence analysis of the Orai1 promoter revealed several palindromic, dihexameric motifs, also known as androgen-responsive elements, that identify androgen receptor binding sites [80]. These results indicate that downregulated androgen receptors ultimately deregulate Orai1 in the aggressive, androgen-independent stage of prostate cancer, resulting in decreased SOCE and increased apoptotic resistance. It remains to be resolved whether, in androgen-independent prostate cancer, the Ca^2+^ crosstalk between ER and mitochondria and/or balance between pro- and anti-apoptotic factors is altered and thus determines apoptosis resistance. Nevertheless, prostate cancer cells benefit from the abrogated SOCE, whereas upregulated SOCE has a pro-survival and pro-migration effect in several other cancer cell types [53,82,108,248,257,272,275,276]. Although these two scenarios are opposing, they both represent a major advantage for cancer cells and leave room for further investigations. Among other CRAC channel components, the downregulation of STIM2 contributes to apoptotic resistance of HT29 colorectal cancer cells, whereas TRPC1 and Orai1 expression was enhanced [254]. Moreover, Orai3 plays a role in apoptotic resistance in breast cancer cells. There, it controls the expression of the p53 protein via the pro-survival PI3K pathway [253]. These findings again suggest the potential for isoform-specific treatments (Figure 4B). Interestingly, the impact of CRAC channels on apoptosis has been reported only for a few cancer types. The detailed mechanisms of the effects on apoptotic pathways remain to be explored.

#### 2.2.4. Epithelial–Mesenchymal Transition (EMT), Migration, and Invasion

CRAC channel components are further of great importance in EMT, migration, and invasion of tumor cells [82,127,255,263,278,280]. In most cancer types, including breast [82], gastric [53], glioblastoma [88,256], melanoma [108,276], and renal [277] cancer, migration and metastasis have been identified to be controlled by both STIM1 and Orai1. Interestingly, in cervical [248], colorectal [85], and liver [257] cancer, only STIM1 has been detected to be critical, and in esophageal [255] cancer, only Orai1 has so far been detected as such. Additionally, Orai2 in the hematologic tumor type, acute myeloid leukemia cells (AMLC) [245], and STIM2 in melanoma [108] manifest the invasive phenotype of the respective cancer types [53,81,106,257,272,280].

The expression of important markers of EMT and requisite regulators of mesenchymal cell migration, such as vimentin or fibronectin, is downregulated in esophageal [255] and gastric [53] cancer upon STIM1 and/or Orai1 silencing. The transition to a more motile and invasive tumor type was accompanied by a loss of E-cadherin function, which was upregulated upon the knock-down of Orai1 in esophageal [255] and gastric cancer [53].

Among factors that promote cell migration, the expression level of small GTPases Ras or Rac is critical in STIM1- and/or Orai1-dependent cancer types, as reported in breast [82] and esophageal cancer cells [255]. Moreover, the function of the Pyk2 kinases and the protease calpain, which control cytosolic scaffold proteins (e.g., α-spectrin) and thus focal adhesion dynamics, is modulated by STIM1 in migratory cervical cancer cells [248]. Indeed, silencing STIM1 attenuates invasive migration of cervical cancer cells, whereas overexpressed STIM1 enhances it [248]. In glioblastoma cells, Orai1 expression is upregulated and linked to their enhanced invasion by controlling the phosphorylation of kinase Pyk2 [272]. CRAC channel proteins are further involved in the regulation of cancer cell migration by controlling the expression of FAK through phosphorylation. Whereas in AMLC, Orai1 and Orai2 control the expression of FAK to promote focal adhesion formation [261], in liver cancer, STIM1 is critical for FAK dephosphorylation to initiate detachment of focal adhesions [256], indicating the important potential of isoform-specific therapeutic targets (Figure 4C).

The reversal of EMT in a highly aggressive type of breast cancer depends on the enhanced expression of SPCA2, which triggers constitutive Ca^2+^ influx via Orai1 [273] (Figure 4C). This potentially occurs via Ca^2+^-dependent Wnt signaling. Specifically, increased Ca^2+^ levels phosphorylate CaMKII and ß-catenin, thereby switching off Wnt and subsequently inhibiting EMT [273]. Moreover, increased SPCA2 levels correlate with decreased vimentin expression, an important mesenchymal marker [273].

In colorectal cancer, it is exceptional that the progression is governed by the pro-inflammatory enzyme cyclooxygenase-2 (COX-2) playing a role in the prostaglandin (PGE2, prostaglandin E2) synthesis. It is activated by the Ras/ERK pathway [23,281,282,283,284,285] and has been also shown to require Ca^2+^ entry and the subsequent activation of transcription factors, NFAT, and CREB [284]. Enhanced production of PGE2 as well as basal and EGF-induced COX-2 expression trigger STIM1-dependent migration. Blocking COX-2 by chemical modulation inhibited colorectal cancer cell migration. Ectopic expression of COX2 and PGE2 sufficiently rescued the effect of STIM1 knock-down, indicating that colorectal cancer cell migration mediated by STIM1 originates from its regulation of COX-2 expression and subsequent PGE2 synthesis. Upregulated expression of STIM1 in colorectal cancer cells induced EMT, whereas STIM1 knock-down showed the opposite effect [86] (Figure 4C). Interestingly, STIM1 was determined to be a direct target of miR-185, a microRNA (miRNA), in colorectal cancer tissues and cell lines.

At this point, the effectors (Pyk2, Ras) by which STIM1 and COX-2/PGE2 affect EMT (vimentin, E-cadherin) as well as migration have not been explored in detail.

#### 2.2.5. Hypoxia Linked to CRAC Channel Components in Cancer

Hypoxia-dependent enhancements in Ca^2+^ levels can underlie the upregulation of Orai channels; however, the mechanism underlying their upregulation remains unknown. Regarding Orai1, it has been shown to upregulate in breast, colon, and glioma cancer cells due to hypoxia together with TRPC6 [63]. This enhanced expression is associated with the Notch pathway, and silencing or inhibition of Notch1 led to reduced Orai1 and TRPC6 expression. In breast cancer cells, enhanced Ca^2+^ levels due to hypoxia were diminished by the application of SKF-96365, a CRAC channel blocker [62,63,64,286,287]. In various breast cancer cells, hypoxia increased Orai3 expression, though the expression levels of other Orai isoforms were unaffected. Remarkably, silencing Orai3 under these conditions failed to reduce SOCE, suggesting that hypoxia is not the only pathway for the upregulation of Orai3 [62] (Figure 4D). An alternative mechanism of positive regulation of Orai3 expression might underlie the action of certain microRNAs (miR), miR18a and miR18b, as recently demonstrated [125]. In another study, the upregulation of Orai3 together with TRPC1 in breast cancer has been linked to HIF-1α, as its silencing reduced the expression of both channels [62,286]. Moreover, hypoxia is capable of inducing EMT [56]. In colon cancer progression, hypoxia augmented Orai1 and Orai3 expression as well as SOCE by increasing the expression of the hypoxia-dependent transcription factor HIF-1/2 [275].

In summary, the role of CRAC channel components in carcinogenesis is multifaceted and depends on the type of cancer, the affected cancer features, as well as the impaired signaling pathway. On the one hand, this opens up targeted therapy opportunities. In particular, STIM2 or Orai3 may function as a therapeutic target for selective cancer therapy. Nevertheless, a more comprehensive understanding of the respective impaired signaling pathways/proteins is still needed to create a more cohesive picture of the interaction between CRAC channel components and cancer signaling pathways.

### 2.3. Cancer Associated with Mutations of CRAC Channel Proteins

Several mutations in STIM1 or Orai1 have been associated with cancers such as colorectal tumor (Orai1 A137V) [283], stomach carcinoma (Orai1 M139V) [288], uterine carcinoma (Orai1 S159L) [289], glioblastoma (Orai1 G183D, STIM1 S116N) [290], neck carcinoma (Orai1 G247S) [291], lung adenocarcinoma (STIM1 A79T, E87Q, W350L, G446C/V) [144], and skin melanoma (STIM1 T517I, S521L) [292] (Figure 5).

All these cancer-related Orai1 mutants have been reported to lead to constitutive activity independent of STIM1 [290]. This likely triggers abnormal Ca^2+^ levels, which might be responsible for the cancer development. Mechanistically, most of these mutations (Orai1 A137, M139, G183, G247) are located at key checkpoints in the Orai TM domains controlling pore opening. Their manipulation likely initiates a global conformational change that finally induces pore opening [182]. Position S159 in Orai1 is located in the loop2 region, which is essential in transferring the gating signal of STIM1 to the pore [293] and controlling the inactivation of STIM1/Orai1 currents [146,294]. However, further studies are still required to understand why this mutation triggers constitutive activity, as it is not directly located in the TM domains.

Among the STIM1 mutants identified to occur in cancer, several (STIM1 H72R, D76V, D78G, A79T, N80K, E87Q, L92P [144]) are located in the canonical EF hand and the hydrophobic cleft and have been found to trigger constitutive activation. Specifically, they lead to constitutive STIM1 cluster formation and consequently to constitutive Ca^2+^ influx through Orai1 channels. The mechanistic role of S116N, located in the non-canonical EF hand, is still unknown. Several other cancer-related mutants are located in the STIM1 C-terminus, which is essential in both maintaining the quiescent state and establishing CRAC channel activation. For instance, W350L is located in the CC2 of STIM1, part of the minimal region sufficient to activate Orai1, and G446C/V, T517I, and S521L are located in the flexible portion of the STIM1 C-terminus. Further studies are still required to understand their mechanistic impact.

### 2.4. Therapeutic Approaches Associated with the Expression of CRAC Channel Components in Cancer

The altered expression of Ca^2+^ ion channels and their interconnectedness with cancer-promoting signaling pathways has opened new possibilities for future therapeutic strategies.

A variety of CRAC channel drugs are available, which has provided valuable insights into CRAC channel mechanisms, but only a few have reached clinical trials due to low selectivity or undesirable side effects. In the treatment of cancer, SKF-96365, a SOCE/Orai1 blocker, has been shown to impair cell proliferation of GBM cells in vitro [256] and metastasis of breast cancer cells in vivo [295]. The CRAC channel inhibitor Diethylstilbestrol (DES), a synthetic ethinyl estrogen, is a possible AR-independent prostate cancer treatment. Hypothetically, DES binds to Orai via steroid-binding sites or even affects channel properties [55]. Synta66, which selectively inhibits Orai1 [296,297], prevents SOCE in three glioblastoma cell lines (U-87 MG, LN-18, A172). Interestingly, it does not affect their division, viability, and migration [298].

Carboxyamidotriazole (CAI) interferes indirectly with SOCE via the PI3K/Akt pathway. Tested as a potential therapeutic, it affects the expression of BCL2 members [299,300] in ovarian cancer [301,302]. Therefore, the inhibitory effect of CAI on SOCE in ovarian cancer could be due to both mitochondrial Ca^2+^ overload and inhibition of Ca^2+^-dependent survival pathways [55].

Drebrin, an actin-reorganizing protein boosting SOCE, is linked to prostate cancer cell invasion, especially at earlier stages of cancer development. There is evidence that BTP2, an Orai1 inhibitor, targets drebrin [303]. Notably, among pyrazoles, BTP2/Pyr2 and Pyr3 can block Orai1, TRPC3, and STIM1 and inhibit melanoma [276,304].

Additionally, store-operated channels in acute myeloid leukemia cells containing Orai3 have been reported to be targeted by tipifarnib, a farnesyltransferase inhibitor, that prevents farnesylation of Ras [305]. This enhanced cytosolic Ca^2+^ levels through Orai3 and caused cell death of AMLC lines [262].

RP4010, a synergistic drug, prevents SOCE activation by inhibiting Orai1. The drug is undergoing clinical trials and might serve as a treatment for pancreatic ductal adenocarcinoma [306] and esophageal cancer [307].

In addition to their function as drug targets, CRAC channel proteins can facilitate the circumvention of a therapeutic effect, known as therapy resistance. This is particularly challenging in the case of the chemotherapeutic agents 5-fluorouracil (5-FU) and cisplatin, that induce autophagy and cell death in cancer cells. In hepatocarcinoma tissues, overexpression of Orai1 impairs the effect of 5-FU [251], making Orai1 an indicator of hepatocarcinoma sensitivity to 5-FU. Similarly, 5-FU treatment in pancreatic cancer significantly increased STIM1 and Orai1 expression, impeding cell death [265]. In NSCLC cells, a blockade of SOCE or STIM1 silencing enhanced cisplatin-induced apoptosis, and STIM1 overexpression reduced apoptosis [84]. Supportively, cisplatin-treated cells revealed downregulated STIM1 expression. In another study [308], the expression of Orai3, but not Orai1, has been shown to lead to cisplatin resistance in bronchial biopsies. A shift in the Orai1:Orai3 expression ratio increases SOCE, as well as the levels of cancer stem cell markers, a mechanism potentially linked to the PI3K/Akt pathway [308].

In summary, although several studies have reported the effect of CRAC channel drugs on various cancers, several questions remain to be addressed, such as why cancer cells continue to proliferate and migrate even though the applied CRAC channel drug blocks store-operated currents. Moreover, of the selective CRAC channel drugs currently available, only a few have made it to clinical trials [309]. Thus, there is a need for targeted therapeutic approaches and novel selective drugs. However, because CRAC channels are ubiquitous, targeting them with highly selective agents may still lead to undesirable side effects. Thus, it is important to focus on cancer type-specific deregulated pathways in therapy development, such as the unique SPCA2-Orai1 or SK3-Orai1 co-regulation in breast cancer or the specific role of Orai3 in various cancer types.

## 3. The Range of Ca^2+^-Activated Ion Channels

An important property of many ion channels is not only the transport of Ca^2+^, but other channels also sense Ca^2+^, either by direct binding of Ca^2+^ or indirectly by Ca^2+^-binding proteins, such as CaM [310,311,312]. On Ca^2+^ channels, Ca^2+^ often has an inactivating effect, providing a negative feedback mechanism that protects against too much Ca^2+^ entering the cell [114]. Other ion channel types have evolved such that their opening and closing are regulated by Ca^2+^, such as Ca^2+^-activated K^+^ channels and Ca^2+^-activated Cl^-^ channels [105,260,313,314,315,316,317,318,319,320,321,322,323]. Several members of both channel families are crucial in the proliferation and migration of different cancer types [319,320,321,322,323,324,325,326,327,328,329,330,331]. In the following, we will focus on the structure/function relationship of the Ca^2+^-activated K^+^ channels and their role in cancer.

### 3.1. Ca^2+^-Activated K^+^ Channels

Ca^2+^-activated K^+^ channels include the large (BK), intermediate (IK or SK4), and small conductance Ca^2+^-activated K^+^ channels (K_Ca2+_: SK1, SK2, and SK3) [332,333], whereas the focus here is on SK channels. Structurally, SK channels are comparable to voltage-gated K^+^ channels, containing six TM regions (S1–S6) with the pore region located between TM5 and TM6 and both the N- and C-termini located at the cytosolic side [315] (Figure 6A). However, they lack the voltage sensor in TM4, leading to their voltage independence [312,317]. SK1, SK2, and SK3 channels possess a conductance of 2-20 pS, whereas for SK4 channels, it is in the range of 20-85 pS [334]. The activation of these SK channels results in hyperpolarization of the membrane potential [332,333]. SK1-3 occurs mainly in the nervous system, whereas SK4 channels are predominantly expressed in epithelial and blood cells and in some peripheral neurons [315,335].

Two cryo-EM structures of the closed and open state of SK4 are currently available [312]. They can be also assumed for the other SK isoforms due to the high sequence similarity between the four SK isoforms. These structural resolutions confirm the tetrameric conformation. The pore formed by a re-entrant loop between S5 and S6 is surrounded by S1 and S4 helices. The S4 and S5 helices are connected by the so-called S4–S5 linker, which consists of two α-helices, S_45_A and S_45_B [312] (Figure 6B,C).

SK ion channels are activated by changes in intracellular Ca^2+^ levels [310,311,315,332]. An increase in cytosolic Ca^2+^ concentration activates the SK channel, which triggers K^+^ efflux from the cell due to the K^+^ concentration gradient with 140 mM K^+^ inside and only 5 mM K^+^ outside [325]. However, SK channels are not able to sense the intracellular Ca^2+^ concentration directly, but additionally possess the Ca^2+^-binding protein CaM constitutively bound to the calmodulin-binding domain (CaMBD) in the C-terminus of each α-subunit. Functional studies of SK channels have demonstrated that the CaM C-lobe is constitutively bound to the SK channel, and the N-lobe is bound upon cytosolic Ca^2+^ elevations [310,311,312,337]. This mechanism has later been confirmed by the SK4 cryo-EM structure, which predicts that four CaM bind to a channel tetramer [312] (Figure 6B). The dynamic and Ca^2+^-dependent interaction of the CaM N-lobe is formed with the S4–S5 linker of the SK channel and triggers a conformational change within the channel complex to induce pore opening [311,312,315,338]. Specifically, CaM interacts with the S_45_A, which then moves to the cytosolic side. Additionally, S_45_B moves away from the pore region, leading to further changes in S6, finally causing channel activation [310] (Figure 6C).

In summary, activation of the SK channel by constitutively bound CaM occurs upon an increase in intracellular Ca^2+^ levels. CaM-Ca^2+^ binding results in altered binding of CaM and triggers a conformational change of the SK channel that causes pore opening.

### 3.2. SK Channels and Cholesterol-Rich Regions

Because the interplay between SK3 and Orai1 that triggers breast cancer cell migration occurs in cholesterol-rich regions, we present here recent findings on the individual regulation of SK channels by lipids [292,316,325,329,339,340], specifically, PIP_2_ and cholesterol.

CaM-dependent activation of SK channels is modulated by PIP_2_, as PIP_2_-depletion inhibits SK2 channels. The PIP_2_ binding site has been identified at the SK–CaM binding interface. CaM phosphorylation causes a change in the interaction of amino acids at the PIP_2_ binding site and reduces the affinity of SK2 for PIP_2_ [340].

Regarding cholesterol-dependent modulation of Ca^2+^-activated K^+^ channels, it is known that cholesterol-mediated regulation of SK channels is dependent on Cav-1. In contrast, the function of BK and IK channels is regulated by cholesterol independently of Cav-1. Cholesterol inhibits BK channels, possibly due to an altered open probability, but not a change in the unitary conductance [341]. Moreover, the alkyl-ether-lipid Ohmline, which reduces SK3 channel activity, gives a hint of cholesterol-dependent SK3 channel regulation [325,329]. Ohmline triggers membrane disordering with increasing cholesterol levels. Molecular dynamics simulations have shown that Ohmline interacts with the carbonyl and phosphate groups of sphingomyelin and stearoylphosphatidylcholine and, to a lesser extent, with cholesterol. It has therefore been suggested that Ohmline removes cholesterol–OH groups from their major binding sites and forces a new rearrangement with other lipid groups. This leads to membrane restructuring and disorder, which could be a possible explanation for Ohmline-induced inhibition of SK3 channels [339]. Despite these findings, the detailed molecular mechanisms of cholesterol-mediated modulation of K_Ca2+_ channels are still unclear.

### 3.3. SK Channels and Cancer

SK channels, like CRAC channels, are involved in the development of diverse cancer hallmarks. Gene expression of SK channels was detected in breast cancer (SK2 and SK3) [327], glioma (SK2) [324], medulloblastoma (SK3) [328], melanoma (SK2 and SK3) [342], colon (SK3), and prostate cancer (SK3) [331,343,344,345]. Interestingly, the detected SK transcripts do not necessarily confirm the expression of the functional protein, as reported for the SK2 gene in breast cancer and glioma [324,327]. Transcripts of SK1 have been only detected in tumor biopsies; however, there is no evidence of their functional role there [324,346]. Although it remains unresolved whether ion channels contribute to cell transformation or are a product thereof, targeting ectopic expression of SK channels in a given tissue may offer both prognostic and therapeutic opportunities.

SK4 is expressed in different cancer types, including glioma [347], glioblastoma [348], breast [349], prostate [350], lung [351], hematologic [352], melanoma [353], colorectal [354], renal carcinoma [355], brain tumors [326], pancreas [356], and papillary thyroid [357], thereby controlling cancer hallmarks [358,359,360]. For instance, in primary breast cancer cells [360,361] and various breast cancer cell lines [358], high SK4 expression levels have been found in line with the electrophysiological evidence obtained upon the application of the SK4 activator, TRAM-34. In MDA-MB-231 cells, the suppression of SK4 channels significantly blocks cell proliferation and migration and elevated apoptosis. In colorectal cancer, SK4 contributes to cell migration and invasion, which is related to the dysfunction of proteins of the RAS/ERK pathway (KRAS), hypoxia (HIF1α), and intracellular ROS production [359] (Table 2).

The catalogue of somatic mutations in cancer (COSMIC) [365] reports over 642, 1398, 1557, and 371 entries for the SK1, SK2, SK3, and SK4 channels, respectively. However, these entries typically represent the outcome of whole exome and RNA sequencing. To date, no cancer-related mutants of a single SK channel family member, but only disease-related SK3 mutations linked to Zimmermann–Laband Syndrome, were functionally characterized [366].

The development of cancer features triggered by SK channels generally occurs in a Ca^2+^-dependent manner. Specifically, an increase in intracellular Ca^2+^ levels in SK channel-expressing cells leads to their activation and thus to potassium (K^+^) efflux, which in turn promotes Ca^2+^ entry via Ca^2+^ channels in a positive feedback loop. The increase in intracellular Ca^2+^ levels achieved in this manner controls Ca^2+^-dependent cancer hallmarks [105,292,325], as we outline in detail in chapter 4 (Table 2).

#### 3.3.1. Proliferation

Insights into the influence of the SK channel on cell cycle progression have only been gained for SK4. It is able to indirectly control cell cycle progression by driving cellular Ca^2+^ entry, as inhibition of SK4 led cells to accumulate in the G1 phase, whereas the number of cells in the cell cycle S phase decreased [367,368]. In primary breast tumors, SK4 is essential for growth factor-dependent Ca^2+^ entry, cell cycle progression, and the proliferation rate of primary breast tumor cells [360] (Figure 7). Obviously, there are only a few studies on the role of SK channels in regulating cell cycle events in cancer cells. Further studies are still needed to understand the molecular mechanisms of SK channels in cancer cell proliferation, in particular, to resolve which cell cycle effectors play a crucial role. 

#### 3.3.2. EMT, Migration, and Metastasis

In particular, the SK3 channel contributes to cell migration and metastases (Figure 7). Supportively, siRNA knock-down of SK3 prevented the migration of MDA-MB-435s cells, in accord with their high metastatic potential [327]. Remarkably, other cancerous (MCF-7) and non-cancerous (184A1) breast cell lines, which physiologically lack SK3 expression, increased their migration potential upon induced SK3 expression. Analogously, SK3 is critical for the migration of melanoma cells, as treatment with apamin significantly reduced the migration of certain melanoma cell lines (518A2, Bris, HBL) [342]. Knock-down of SK3 by shRNA abolished 518A2 cell migration, whereas conversely, overexpression of SK3 in non-SK3-expressing cells increased the migration capacity. Additionally, melanoma cell motility has been demonstrated to decrease as a consequence of PM depolarization promoted by increased extracellular K^+^ concentration. Furthermore, 2D and 3D motility assays suggest that the migration of melanoma cells depends on SK3 activity [342].

SK4 governs oncogenic pathways controlling migration and metastasis. SK4 mRNA expression is enhanced in MDA-MB-231 cells upon the application of growth factors (e.g., EGF) that trigger EMT. Indeed, silencing of SK4 expression abolished the expression of certain EMT markers (e.g., Vimentin) [358] (Figure 7).

Overall, there is evidence that SK channels are present in cancer cells and can control the development of cancer features. However, knowledge about the influence of SK channels on individual steps in signaling cascades regulating the cell cycle or migration is limited. In this context, identification of the effects on key players modulating, for instance, the focal adhesion dynamics (PYK-2, Rac1) is required. A detailed understanding in this regard could improve therapeutic options for cancer cells.

### 3.4. Therapeutic Approaches Associated with the Expression of SK Channels in Cancer

Drugs targeting SK channels include peptides extracted from scorpions, sea anemone or bee venom, synthetic analogues, and chemically synthesized modulators (e.g., Cyppa, 1-EBIO, NS8593, NS306, TRAM34). However, all of these modulators arose as limited in medical use due to their high toxicity or low efficiency. Among the blockers, particularly those targeting SK4 have been suggested as promising for the treatment of various cancers [325]. Specifically, this compound blocked cell cycle progression in murine breast cancer cells [360], reduced the mass of human endometrial cancer cells (HEC-1-A) [369] and non-small-cell lung cancer cells (A549-3R) [351], or impaired infiltration of glioblastoma [370].

To overcome therapy resistance [371], SK3 and SK4 represent promising candidates. Supportively, the SK3 channel gene is one of 1298 genes contributing to ovarian cancer drug resistance [372]. Notably, in bortezomib-resistant BN myeloma cell lines, 16-fold upregulated gene expression of the SK3 channel has been reported, reflecting its relevance to drug resistance [373]. Cisplatin-resistance of human epidermoid cancer cells is regulated by SK4, due to control of proliferation and regulation of cell volume. Activation of SK4 using 1-EBIO or SKA-31 boosted apoptosis in cisplatin-resistant cancer cells [354,374]. Overexpression of SK4 in breast cancer cells increased resistance to the chemotherapeutic agent gemcitabine by upregulating an anti-apoptotic BCL2-protein [375].

The alkyl-ether lipids edelfosine and Ohmline have emerged as promising lipid-antimetastatic agents. Both block SK3-dependent cancer cell migration and metastasis, likely due to lipid membrane reorganization, which inhibits SK3 channels [325]. Edelfosine, for instance, effectively blocks migration and invasion of urothelial carcinoma cells expressing SK3 [376]. Hypoxia-induced upregulation of SK3 together with the EMT transcription factor Zeb1 in prostate cancer cells is promoted by the anti-androgen Enzalutamide, used for the treatment of castration-resistant prostate cancer [331,377]. Enzalutamide is known to dysregulate Ca^2+^ signaling, which is crucial for EMT progression, contributing to therapeutic escape [331]. Ohmline could prevent hypoxia-triggered EMT pathways [377] and Enzalutamide-dependent Zeb1 expression [331]. Moreover, Ohmline has been reported to interfere with the co-regulation of SK3 and Orai1 that is crucial in breast, colon, and prostate cancer progression (see Section 5) [320,323].

Overall, alkyl ether lipids appear to be the most promising and selective drug candidates for the treatment of SK3-dependent cancers.

## 4. Co-Regulation of Ca^2+^ and K^+^ Ion Channels Linked to Cancer Development

In addition to their individual role, K^+^ channels interact with Ca^2+^ ion channels to control various cellular functions, including also the fine-tuning of the intracellular Ca^2+^ concentration. Voltage-gated K^+^ (K_v_) and K_Ca2+_ channels have been classified as Ca^2+^ amplifiers because of their interplay with Ca^2+^ channels [321,322,327,360]. These K^+^ channels, when activated, induce membrane hyperpolarization either by increased intracellular Ca^2+^ levels in the case of K_Ca2+_ channels or by membrane depolarization in the case of K_v_ channels, thereby enhancing the driving force for Ca^2+^ influx. Several studies indicate that co-regulation of K^+^ (K_v_, K_Ca2+_) and Ca^2+^ channels (P2X7 receptor, CRAC, TRP, voltage-gated Ca^2+^ (Ca_v_) channels) plays a significant role in cancer cell progression, growth, proliferation, and invasion/migration [320,321,322,327,360]. Specifically, the SK3 channel, together with the P2X7 receptor, contributes to the cysteine cathepsin-dependent invasiveness of breast cancer cells [330] (Table 2). Moreover, BK channels have been shown to form molecular complexes with Ca_V_3.2, promoting the proliferation of prostate cancer cells [378]. SK4 (IK) channels have been found to co-immunoprecipitate with TRPV6 in LNCaP cells [319]. Human Ether a-gogo potassium Channel 1 (hEag1) associates with Orai1 and regulates breast cancer cell migration through Orai1-dependent Ca^2+^ entry [379,380]. The activity and membrane trafficking of these proteins is mediated by SPCA2 promoting a basal Ca^2+^ influx through store-independent (SICE) Orai1 activation. This trio complex triggers collagen I-induced proliferation and survival of breast cancer cells [381]. The most extensively studied interplay of K_Ca2+_ channels, in terms of structural and functional requirements and their role in cancer, represents that of Orai1 and SK3 [105,292].

### 4.1. SK3-Orai1 Interplay

Besides the critical role of the physical interaction of SK3 and Orai1 in healthy body functions, as evidenced by its beneficial role in avoiding excessive smooth muscle contraction in the gallbladder smooth muscle of guinea pigs [382], this positive feedback loop contributes to the development of cancer hallmarks (Figure 8). Specifically, breast and colon cancer progression has been demonstrated to be governed by the interplay of SK3 and Orai1. Though both cancer types exhibit SK3 and Orai1 expression [257,311], healthy tissue does not express SK3 channels [320,327,342]. Co-localization of SK3 with Orai1 has been detected via co-immunoprecipitation in breast and colon cancer cells [327,329].

Despite the interplay between SK3 and Orai1 being extensively studied in breast and colon cancer cells (see Section 4.1.2), the molecular determinants required for their co-regulation are poorly characterized. In the following, we describe our recently uncovered key determinants that are essential for their SK3-Orai1 co-regulation.

#### 4.1.1. The Molecular Determinants of the SK3-Orai1 Interplay

We discovered that the interplay between Orai1 and SK3, similar to that in breast and colon cancer cells, also occurs in HEK 293 cells [343]. We demonstrated that the co-expression of Orai1 and SK3 led to significantly higher K^+^ currents than in the absence of Orai1. This was accompanied by their close co-localization and co-immunoprecipitation, but a direct interaction by FRET could not be detected so far. The inhibition of K^+^ currents by an inhibitor specific for SK3 channels left tiny inwardly rectifying Ca^2+^ currents in SK3 and Orai1 co-expressing cells, which could be blocked by La^3+^. In contrast, K^+^ currents in SK3-only expressing cells were completely blocked by the SK3 inhibitor (NS8593). This suggests, in agreement with previous findings in cancer cells [320,321,322,323], that the co-occurrence of Orai1 and SK3 increases cytosolic Ca^2+^ levels specifically through Orai1, which together with SK3 triggers the positive feedback mechanisms [105,325]. Intriguingly, we did not detect enhanced Ca^2+^ entry in SK3-Orai1 expressing cells, and NFAT translocation was only marginally enhanced. This indicates that already, very low local alterations in Ca^2+^ levels are sufficient to trigger the SK3–Orai1 interplay. Moreover, we found that K^+^ currents in SK3 and Orai1 co-expressing cells could be reduced to the level of SK3-expressing cells not only when the general Ca^2+^ channel blocker La^3+^ was applied, but also the CRAC channel-specific inhibitor GSK-7975A [343]. Co-expression of Orai1 E106Q with SK3 also resulted in significantly reduced K^+^ currents compared to the presence of wild-type Orai1, despite their co-localization being maintained (Figure 8A). Moreover, SK3 K^+^ currents in the presence of Orai1 gradually enhanced with increasing extracellular Ca^2+^ concentrations and were abolished in the presence of 0 mM Ca^2+^ or divalent-free Na^+^ extracellularly, proving that Ca^2+^ from the extracellular space controls SK3 channel activity. Decreasing intracellular Ca^2+^ levels using intracellular EGTA strongly reduced (300-500 µM) or abolished (1000 µM EGTA) SK3 K^+^ currents, both in the absence and presence of STIM1 (Figure 8A). Overall, our results show that SK3 K^+^ currents are specifically enhanced by Ca^2+^ influx through Orai1 [343].

In addition to Orai1-mediated enhancement of SK3 K^+^ currents, CaM is known to act as the Ca^2+^ sensor that typically controls SK channel activity. We have shown for SK3, in agreement with previous studies [337,383], that CaM wild-type enhances SK3 K^+^ currents, whereas CaM_MUT_, containing all four mutated EF-hands, completely abolishes these currents [343,383] (Figure 8A). Regarding SK3/Orai1 co-regulation, a triple expression of Orai1, SK3, and CaM did not strongly enhance K^+^ currents compared with those obtained in the absence of Orai1. However, SK3 K^+^ currents completely abolished by CaM_MUT_ could be partially restored by Orai1. Characterization of the FRET of CaM/CaM_MUT_ and SK3 revealed that robust FRET efficiency was reduced in the presence of Orai1. It is plausible that Orai1 sequesters part of CaM. However, since Orai1 can partially restore SK3 function lost due to CaM_MUT_ overexpression, it is possible that Orai1 and CaM proteins compete for an intact SK3–CaM binding site to control SK3 channel activity [343], whereas the nature of the direct or allosteric interplay between Orai1 and SK3 remains to be elucidated.

The molecular determinants within Orai1 that are required for the interplay with SK3 are the N-terminus, except for the first 20 aa, as well as the C-terminus (aa 281–301) and an intact pore geometry, as demonstrated by a set of LoF-Orai1 mutants. They were unable to boost SK3 channel activity without affecting the co-localization with SK3. However, STIM1 coupling to Orai1 or an intact STIM1 coupling site within Orai1 (L273) is not required for intact Orai1-SK3 interplay, as evidenced by the preserved boost of Orai1-mediated SK3 channel activity in CRISPR/Cas9 STIM1/Orai1 DKO HEK 293 cells. In agreement with this, Orai1 L273D was also able to trigger the amplification of SK3 K^+^ currents [343]. It is still an interesting fact that in contrast to STIM1-mediated Orai1 activation, almost the entire N-terminus, particularly the residues between aa20–70, are also required for an intact SK3/Orai1 co-regulation [343]. This region contains an AC8-binding domain (residues 26–34 [384]), two putative protein kinase C (PKC) phosphorylation sites (S27 and S30 [385,386]), a putative PIP2-binding sequence (residues 28–33 [207]), and a caveolin (Cav)-binding region (residues 52–60 [233]). It remains to be tested whether any of these factors play a critical role in the interplay of SK3 and Orai1.

Interestingly, co-expression of STIM1 with Orai1 and SK3 showed an inhibitory effect on K^+^ current amplification under physiological conditions, as STIM1 activation removes Orai1 from SK3, and locally increased Ca^2+^ levels cannot affect SK3 channel activity. Indeed, global Ca^2+^ level increases also enabled the amplification of SK3 K^+^ currents in the presence of STIM1 and Orai1 [343] (Figure 8A). This supports our findings that local, almost undetectable Ca^2+^ elevations are sufficient to govern the interplay of SK3 and Orai1 in the absence of STIM1 [343]. It remains to be determined whether these observations in dependence on STIM1 play a role in native tissue and cancer.

Our findings validate the hypothesis that Orai1 and SK3 are in close proximity to each other and narrow down critical determinants for this, although a direct interaction could not be confirmed. Further studies are still required to uncover potential molecular links manifesting the store-independent and STIM1-independent interplay of Orai1 and SK3. Among the variety of accessory proteins, SigmaR1, which is known to interact with STIM1 and SK3 (see Section 4.1.2), or SPCA2, reported to interplay with Orai1, could act as critical candidates. Additionally, the lipid environment including SK3-Orai1 clusters may initiate and modulate their interplay, but this first requires a detailed understanding of the molecular role of lipids on the individual channels.

#### 4.1.2. The Interplay of SK3 and Orai1 in Breast, Colon, and Prostate Cancer Cells

In cancer cells, the SK3-Orai1 complex is located within cholesterol-rich regions, where it triggers a constitutive Ca^2+^ influx specific to Orai1. This promotes cell migration, a prerequisite for metastases development. This finding is further reinforced by the fact that disruption of cholesterol-rich regions, where Orai1 and SK3 have been shown to form complexes, for example, by Ohmline, leads to loss-of-function and disrupts cancer cell migration [316,320,321,322,323,329]. Whether chemical cholesterol depletion, for instance, by MβCD, has similar effects remains to be investigated. Moreover, the molecular mechanisms and affected markers in how SK3-Orai1 triggers migration are still unknown.

Interestingly, in breast cancer cells, the second member of the CRAC channel, STIM1, does not play a significant role in these processes (Figure 8B), suggesting a distinct pathway of Ca^2+^ signaling in breast tumors [105,320,321,322,323,325]. Currently, only SPCA2 is known to activate Orai1 in a STIM1-independent manner; however, whether it is a critical factor that stabilizes the interplay with SK3 is unknown [278,381,387]. In contrast, colon cancer cell migration is manifested through the formation of a triple channel complex of the TRPC1/Orai1/SK3 channels in cholesterol-rich regions, which is further facilitated by STIM1 [323]. The assembly of this complex is mediated by EGF phosphorylation of STIM1 and induced activation of the Akt pathway. A positive feedback loop exists: (i) EGF and Akt phosphorylation of STIM1 induce SOCE, which promotes cell migration via the formation of Orai1 and TRPC1 complexes within SK3 channel-rich caveolae-lipid rafts; (ii) Akt activation is triggered by SOCE, which is enhanced by SK3 channel-induced membrane hyperpolarization, which increases the electrochemical driving force to permit Ca^2+^ entry and SOCE; (iii) and the Akt pathway promotes the activity of Rac1/Calpain, which amplifies SOCE, and consequently Akt. Simply stated, SOCE triggers activation of the SK3 channel and the Akt pathway, which mediates the activity of Rac1/Calpain, which in turn leads to the amplification of SOCE (Figure 8C; Table 3). Disruption of cholesterol-rich regions by Ohmline led to the disassembly of the channel complex, resulting in decreased Akt phosphorylation. Notably, caveolae-lipid rafts function as platforms for EGFR-activated signaling pathways. In colorectal cancer cells, upregulated EGFR expression has been identified and linked to prognostic outcome [388,389]. Since we observed that STIM1 binding to Orai1 reduces the co-localization with SK3 in HEK293 cells [343], it might be interesting to determine whether and how endogenous as well as overexpressed STIM1 affect the co-localization and interplay of SK3 and Orai1 in colon cancer cells.

A further report [321] indicates that the SK3–Orai1 interplay is promoted by the stress-activated chaperone SigmaR1 in breast and colon cancer cells. Specifically, SigmaR1 physically interacts with the SK3 channel, as shown via confocal microscopy studies, whereas the binding site remains to be narrowed down. Noteworthy, upregulated expression of SigmaR1 associates with prognostic outcome and tumor grade. Upon SigmaR1 knock-down or the application of sigma ligand, igmesine, SK3 currents, and constitutive Ca^2+^ entry was abolished. Remarkably, whereas SigmaR1 knock-down moved both channels to non-raft fractions, the use of igmesine resulted in the dissociation of Orai1 from lipid rafts without disrupting the SK3–SigmaR1 complex. Consequently, the migration of breast and colon cancer cells was impaired [321] (Figure 8B,C). Whether SigmaR1 interacts with Orai1 is currently unknown. Nonetheless, it has been reported that STIM1 can interact with SigmaR1 in HEK 293 cells [390], thus delaying STIM1/Orai1 activation. Whether and how this could affect the Orai1–SK3 co-regulation in cancer cells remains to be investigated.

We recently reported that the SK3 channel is endogenously expressed in cells of the prostate cancer cell line LNCaP, where it regulates proliferation [343]. Interestingly, in electrophysiological and immunoblotting experiments, we detected SK3 channel expression only in passages 3–5 of this cell line, but not earlier or later ones. After the forced expression of Orai1 in LNCaP cells, endogenous SK3 currents were significantly boosted in accordance with our findings in HEK 293 cells [343]. Overexpression of the pore mutant Orai1 E106Q failed to promote SK3 currents, indicating that functional Orai1 is required to increase SK3 channel activity. Moreover, CaM overexpression in LNCaP cells promoted SK3-mediated K^+^ currents, whereas CaM_MUT_ abolished these currents. Remarkably, additional co-expression of Orai1 bypassed the inhibitory effect of CaM_MUT_ [343]. Our results are corroborated by a recent report [331] that showed weak expression of SK3 in LNCaP cells, which could be highly upregulated by Enzalutamide [331,364,377] (Figure 8D; Table 3). It is tempting to speculate that Enzalutamide treatment enforces the co-regulation of SK3/Orai1 in LNCaP cells and allows a more intensive study of key determinants mediating the SK3–Orai1 interplay in this prostate cancer cell line.

## 5. Perspectives

Cell fate is determined by the finely tuned interplay of signaling cascades involving various cellular components such as CRAC and SK channels. However, their dysregulation/-function can be linked to novel biological functions associated with cancer progression. Mechanistically, dysregulation of CRAC and SK channels can lead to altered activity of a number of effectors of distinct signaling pathways, such as CaM/CaMKII/CN [26,28,273,276], PI3K/Akt [26,249,308,323], or Ras/ERK [21,82,359]. Associations between altered expression of CRAC channel components and affected modulators of cell cycle or migration/metastasis have been found in several cancers [53,81,82,108,245,248,257,272,275,276,277]. However, in most cases, the signaling pathways connecting enhanced Ca^2+^ entry to the cancer hallmark(s) are not yet elucidated in detail. For instance, proliferation can be directly regulated by Ca^2+^ via Ca^2+^-sensing proteins or influence transcriptional activity to modulate the expression of cell cycle effectors. Alternatively, PI3K/Akt or RAS/ERK pathways might be affected by either enhanced expression levels of Ca^2+^ ion channels and/or the effect of Ca^2+^ on certain oncogenes in the PI3K/Akt or RAS/ERK pathways. Therefore, it remains open whether Ca^2+^ modulates only a specific path or multiple ones to impact cancer progression in a given cancer type. In most recent studies, the proliferative activity of related cancer cell lines is monitored upon siRNA-mediated inhibition of ion channels in vitro. Nevertheless, evidence of tumor behaviors in vivo is lacking. At this point, the question arises whether the effects of ion channels on proliferation are not merely coincidental because they interact closely with each other, or, possibly with different more relevant components of the Ca^2+^ signaling cascade. Concerning the role of CRAC channel proteins in apoptosis, several have been identified to contribute to pro- or anti-apoptotic mechanisms; however, the detailed effects on certain regulators of cell survival/cell death signaling cascades are not yet fully understood. Key questions include how altered STIM/Orai expression might influence the expression of pro/anti-apoptotic factors, and how mitochondrial/ER Ca^2+^ levels and their interplay are dysregulated. Moreover, the molecular mechanisms of mutations identified to be critical in cancer are only beginning to be deciphered, and their connections to Ca^2+^ dysregulated signaling pathways remain open. In particular, for SK channels as well as their interplay with STIM1/Orai1, the detailed links to various cancer signaling pathways are insufficiently described.

The multifaceted interplay of these ion channels with intracellular signaling pathways is extended by their interaction with each other and with a set of other signaling components, as exemplified by Orai1-SK3 and Orai1-SPCA2 interplay. For such scenarios, the formation of macromolecular complexes appears favorable. For example, SK3 and Orai1 colocalized in cholesterol-rich regions in breast cancer cells [320], the disruption of which abolished SK3-Orai1 interplay, cancer cell migration, and bone metastasis [316,320,321,323,329,391,392]. Nonetheless, the molecular relationships underlying this complex ion channel formation are not well understood. SPCA2-Orai1 or SK3-Orai1 complexes are known to control breast cancer progression. However, whether a triple complex of SPCA2–Orai1–SK3 strengthens their interplay and is crucial for breast cancer progression requires further investigation. Moreover, TRP channels have been reported as critical components controlling cancer progression together with CRAC channel components; however, the detailed mechanistic underpinning of their interplay requires additional studies. Furthermore, it remains open how lipids regulate individual ion channels and their interplay with each other and with accessory proteins. This suggests that pro-survival and pro-apoptotic events are controlled by a complex regulatory network of modulating proteins and cell type-specific signaling pathways.

Altered expression or mutation of CRAC and SK channels is responsible for poor prognosis in cancer patients. Therefore, these channels could serve as suitable prognostic markers as well as attractive targets for cancer therapy, also in an isoform-specific manner. However, of a large number of CRAC and SK channel drugs available, only Ohmline currently has the potential for future clinical use in SK3/Orai1-driven cancer progression. In addition, the expression of CRAC and SK channels increases resistance to chemotherapeutic agents and promotes cell proliferation, necessitating the need to address alternative targets via potential therapeutic strategies.

## 6. Conclusions

Overall, among the various members of the Ca^2+^ signal transduction apparatus, CRAC and SK channels play critical roles in the development of certain cancers. Their modulatory role in Ca^2+^ signaling affects proliferation, apoptosis, and/or migration. Knowledge of these ion channels and their associated cancer features is extensive but insufficient to develop appropriate therapeutic strategies. A more detailed understanding of the mechanisms of ion channels and their connection to cancer signaling pathways will provide new targets for future and more specific therapeutic strategies against cancer.

## Figures and Tables

**Figure 1 cancers-15-00101-f001:**
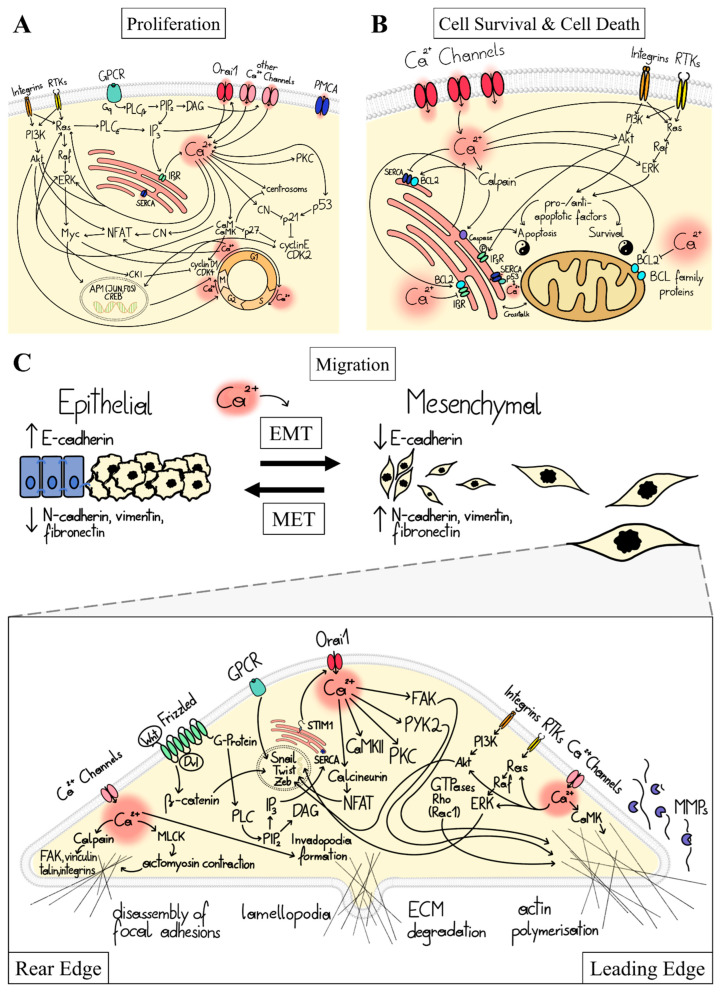
Ca^2+^-dependent cancer signaling pathways controlling proliferation, cell survival, and death and migration. (**A**) The scheme depicts critical pathways for elevation of cytosolic Ca^2+^ levels (via PM ion channels (here, the focus is on the store-operated Ca^2+^ ion channel named the CRAC channel, i.e., thepore-forming component Orai1) or upon ER-Ca^2+^ depletion) that control proliferation at the level of the cell cycle machinery (cyclins, CDKs, CDK inhibitors, centrosome cycle) in a Ca^2+^-dependent manner, either directly via Ca^2+^-binding proteins (CaM, CaMK, CN), or indirectly via transcription factors (NFAT, or immediate early gene (JUN, Myc, and FOS)) and the oncogenic pathways, Ras/ERK and PI3K/Akt. (**B**) Scheme highlights pathways that control cell survival and death factors (e.g., BCL family proteins such as BCL2) in a Ca^2+^-dependent manner. These include factors modulating ER Ca^2+^ uptake and Ca^2+^ crosstalk between ER and mitochondria (SERCA, p53, Akt, IP_3_R, calpains, caspases). Moreover, the Ca^2+^-dependent modulation of cell fate via oncogenic pathways is shown. (**C**) Top: Schematic representation of the transitions between epithelial and mesenchymal states (MET-EMT (mesenchymal-epithelial transition <-> epithelial-mesenchymal transition)) and the characteristic up- and down-regulations of specific gene expression markers (E-/N-cadherin, vimentin, fibronectin), which are controlled by Ca^2+^. Bottom: Ca^2+^-dependent pathways controlling highly coordinated migration together with contacts to the extracellular matrix at the rear and the leading edge of the cell. At the leading edge, Ca^2+^ triggers lamellipodia formation by actin polymerization, the assembly of focal adhesions, the formation of contact with the bottom, and matrix metalloproteinases (involving small GTPase, such as Ras and Rac1, Ca^2+^-dependent factors, including CaMK and proline-rich tyrosine kinase 2 (PYK2); focal adhesion kinase (FAK)). At the rear edge, the cell disconnects from the bottom, involving focal adhesion disassembly (Ca^2+^-sensitive protease calpain, which cleaves FAK, vinculin, talin, and integrins) and actomyosin contraction (MLCK). Software used for all figure drawings: Procreate, Chemdraw (RRID:SCR_016768).

**Figure 2 cancers-15-00101-f002:**
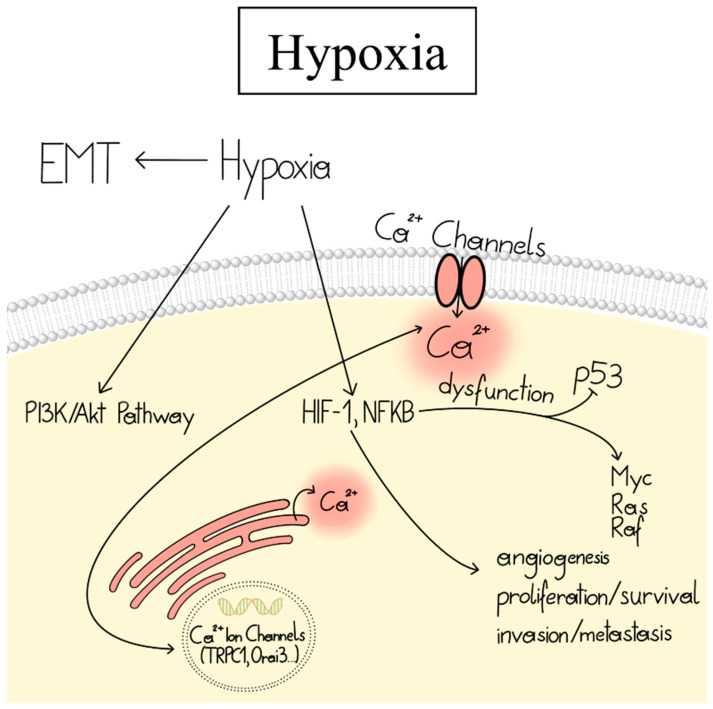
Linkage of hypoxia and the Ca^2+^-signaling machinery in cancer. Hypoxia induces EMT and activates PI3K/Akt, HIF, and NFkB signaling pathways. HIF-1 is involved in angiogenesis, cell proliferation/survival, and invasion/metastasis. It can trigger upregulated expression of Ca^2+^ ion channels, leading to enhanced Ca^2+^ levels that promote transcription of oncogenes. Conversely, Ca^2+^ can modulate HIF-1 signaling. Software used for all figure drawings: Procreate, Chemdraw (RRID:SCR_016768).

**Figure 3 cancers-15-00101-f003:**
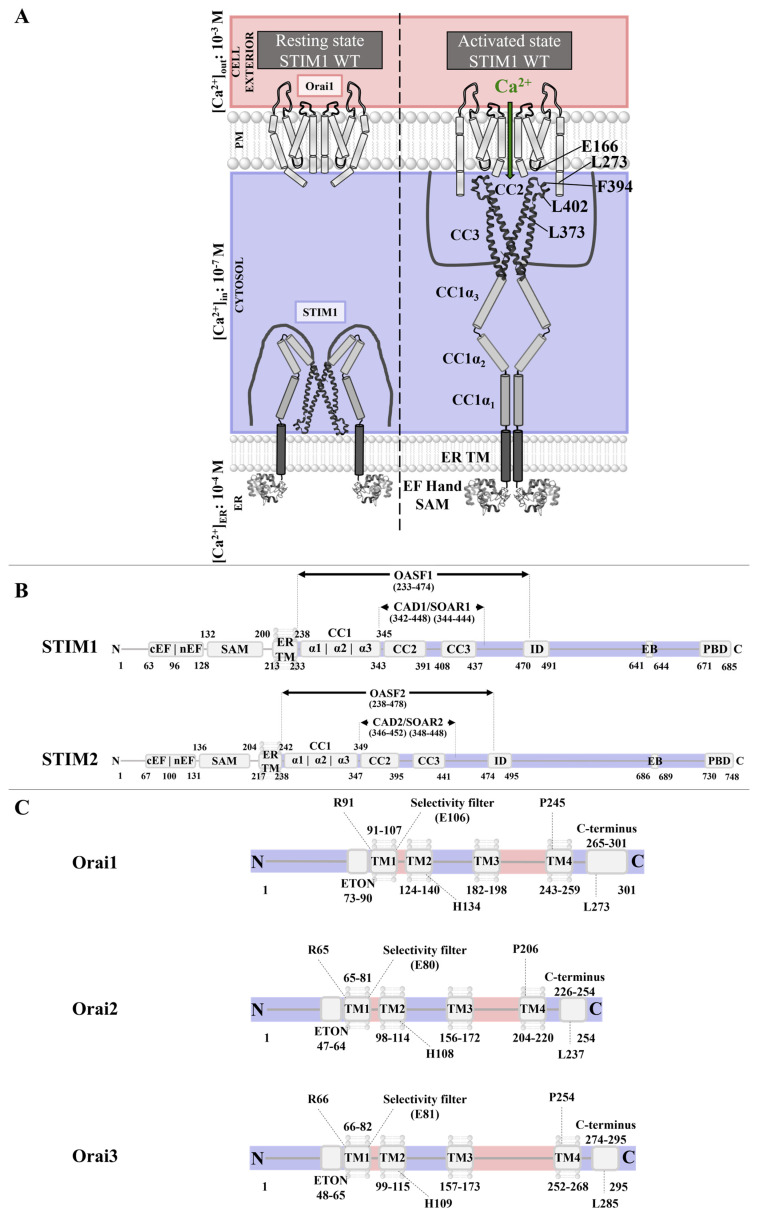
The STIM1-Orai1 activation mechanism and STIM and Orai isoforms. (**A**) Illustration of the activation mechanism of the CRAC channel, starting with inactive STIM1 and Orai1 in the resting state (**left**), followed by the depiction of the two proteins in the activated state allowing Ca^2+^ influx indicated by the green arrow (**right**), with the most important interaction sites highlighted. (**B**) A schematic illustration of the STIM1 and STIM2 isoforms consisting of the respective canonical EF-hand domain (cEF) and the hidden EF-hand domain (hEF), followed by the sterile alpha motif (SAM) in the ER lumen and subsequently the single TM. The C-terminus located in the cytosol contains three coiled-coil domains (CC1–3), more specifically, the three predicted α-helices of CC1 (α1–3). These coiled-coil regions comprise the Orai-activating small fragment (OASF), which spans all three coiled-coil domains, the CRAC-activating domain (CAD), or the STIM1-Orai-activating region (SOAR), which includes CC2 and CC3. The coiled-coil regions are followed by the inactivation domain (ID), the microtubule end-binding domain (EB), and finally, the polybasic domain (PBD) at the very end of the C-terminus. (**C**) An overview of all Orai proteins (Orai1, Orai2, Orai3) highlighting the TM domains alongside major positions for CRAC channel gating (Orai1 aa 72-90—ETON region, R91—SCID mutation position, E106—selectivity filter, H134, P245—critical gating checkpoints, L273—critical STIM1 binding site and analogue sites are labeled in Orai2 and Orai3). Software used for all figure drawings: Procreate, Chemdraw (RRID:SCR_016768).

**Figure 4 cancers-15-00101-f004:**
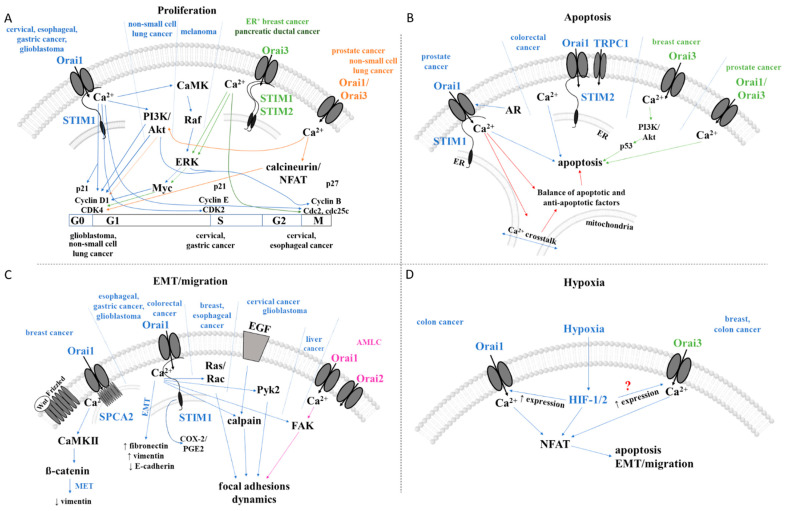
Cancer features and associated CRAC channel-dependent signaling pathways in different cancer types. The schematics summarize the current knowledge of signaling pathways controlling cancer features (proliferation (**A**), apoptosis (**B**), EMT/migration (**C**), and hypoxia (**D**)) of the mentioned cancer types depending on CRAC channel components, TRP channels, or SPCA2. ? … signaling pathway is unknow. Colors indicate which cancer type correlates (arrow in corresponding color) which signaling pathway. Software used for all figure drawings: Procreate, Chemdraw (RRID:SCR_016768).

**Figure 5 cancers-15-00101-f005:**
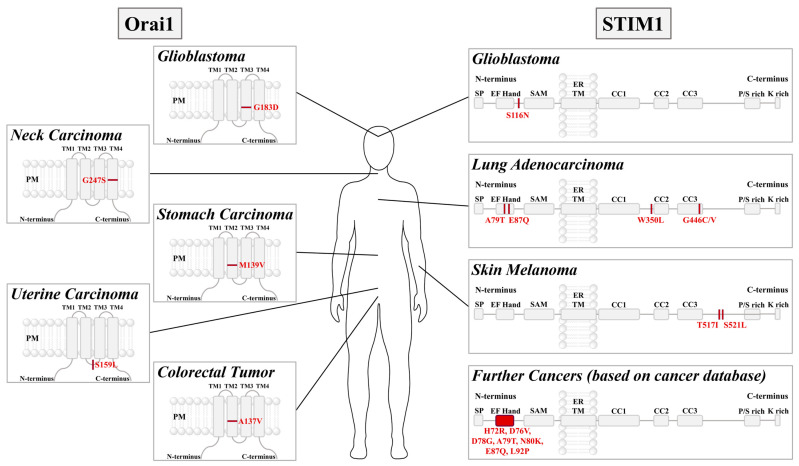
STIM1 and Orai1 cancer-related mutations. Schematic representation of the major components of STIM1 and Orai1 highlighting those point mutations associated with cancer cell development, including glioblastoma, cervical cancer, gastric cancer, uterine cancer, colorectal tumor, glioblastoma, lung adenocarcinoma, skin melanoma, and others [144,290]. Software used for all figure drawings: Procreate, Chemdraw (RRID:SCR_016768).

**Figure 6 cancers-15-00101-f006:**
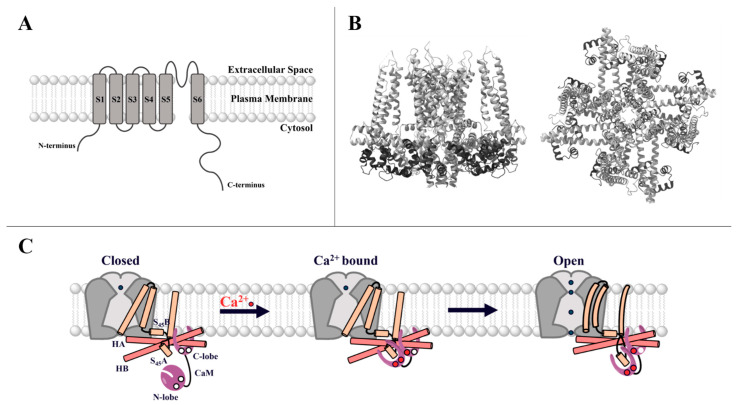
A general overview of SK channels and their activation mechanism. (**A**) Simplified scheme of the 6 TM domains (S1–S6) of the SK channel, with the pore located between S5 and S6 and both the N- as well as the C-terminus in the cytosol. (**B**) Side and top view of SK4 with the constitutively bound CaM in dark gray (PDB ID: 6CNM—visualized by ChimeraX [336]). (**C**) Activation mechanism of the SK channel, starting with an inactive channel and the free CaM N-lobe with no bound Ca^2+^. The C-lobe of CaM is constitutively bound to the channel (**left**). The binding of Ca^2+^, mainly to the CaM N-lobe, triggers a conformational change leading to the interaction between the CaM N-lobe and the S45A helix (**middle**), further communicating a conformational change to the channel, pulling the S45B away from the pore and rendering the channel open (**right**) (adapted from Lee et al., 2018 [312]). Software used for all figure drawings: Procreate, Chemdraw (RRID:SCR_016768).

**Figure 7 cancers-15-00101-f007:**
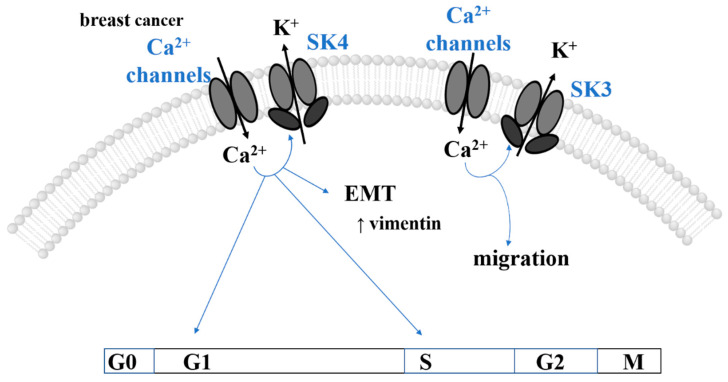
Cancer features and associated SK channel-dependent signaling pathways in different cancer types. The scheme summarizes the current knowledge of signaling pathways controlling cancer features (proliferation and EMT/migration) of the mentioned cancer types depending on SK channels as well as Ca^2+^ signaling. Software used for all figure drawings: Procreate, Chemdraw (RRID:SCR_016768).

**Figure 8 cancers-15-00101-f008:**
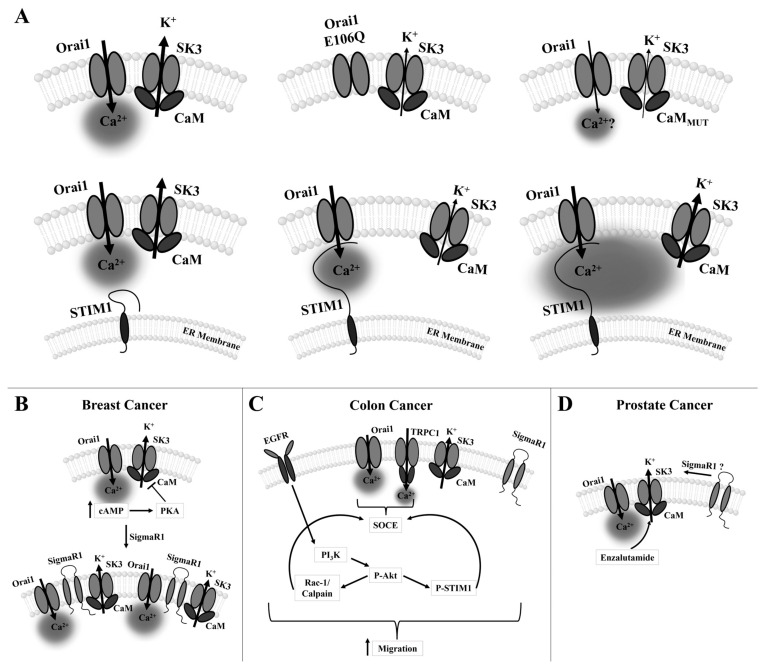
The Orai1/SK3 interplay and methods of its modulation in breast, colon, and prostate cancer. (**A**) Generally, the co-localization of Orai1 and SK3 channels is sufficient for their interplay, which leads to a positive feedback mechanism. Orai1 E106Q pore mutant does not affect its co-localization with SK3 but is unable to boost SK3 K^+^ currents. CaM_MUT,_ which typically abolishes SK3 K^+^ currents, can be rescued by co-localized Orai1. In the presence of STIM1, the SK3–Orai1 interplay remains unaffected in the resting state, and upon store-depletion, STIM1 binds to Orai1 and thus moves SK3 and Orai1 apart from each other. Local enhancements of Ca^2+^ due to STIM1-mediated Orai1 activation do not boost SK3 K^+^ currents, whereas global Ca^2+^ enhancements strongly augment SK3 K^+^ currents. (**B**) In breast cancer cells, the interplay of Orai1 and SK3 boosts metastasis, which can be modulated by the cAMP-PKA pathway (adapted from Clarysse et al. (2014) [322]). Additionally, the stress-activated chaperone SigmaR1 promotes the interplay of Orai1 and SK3 in breast cancer cells (adapted from Gueguinou et al. (2017) [321]). (**C**) In colon cancer cells, migration is promoted by SOCE via a complex of TRPC1, Orai1, and SK3. This involves the activation of the Akt pathway and the phosphorylation of STIM1, leading to a positive feedback loop (adapted from Gueguinou et al. (2016) [323]). (**D**) SK3, reported to be endogenously expressed in the prostate cancer cell line LNCaP, is involved in the regulation of cell proliferation. The application of Enzalutamide enables the upregulation of SK3 expression in LNCaP cells (adapted from Figiel et al. (2019) [364]). Further involvement of accessory proteins, such as SigmaR1, cannot be ruled out and require further investigation. Throughout the figure, arrows indicate the direction of the ion flux. Software used for all figure drawings: Procreate, Chemdraw (RRID:SCR_016768).

**Table 1 cancers-15-00101-t001:** Correlation of cancer types with critical CRAC channel components.

Cancer type		Critical Proteins	Targeted Signaling Pathways	Affected Cancer Hallmarks	Cell Type	Ref
**Breast**		↑STIM1, ↑Orai1	small GTPases ↑Ras and ↑Rac -> ↓focal adhesions ->	migration, metastasis	MDA-MB231	[82]
		↑Orai1 +↑SPCA2	↑vimentin ↑Wnt/Ca^2+^ sig. pathway -> CaMKII -> ß-catenin ->↓Wnt	EMT	MCF-7	[273]
		↑Orai3	↑ER ->	cell proliferation		[126,127]
			↑ERK1/2 ->↑Myc -> ↑cell cycle (G1) ->	cell proliferation		[253]
			↑PI3K -> ↓p53 ->	apoptosis		[271]
			hypoxia ->	EMT	MDA-MB-468	[62]
		STIM2 + Orai1	(↑?)PAR-2 ->	survival, invasion, cancer prognosis	MCF-7, MDA-MB-231	[250]
**Cervical**		↑STIM1, ↑Orai1	↑cell cycle (G1/S (CDK2, cyclin E)) ->	proliferation	SiHa, HeLa, U2OS	[274]
			↑EGF -> ↑calpain ->↑α-spectrin ->	migration, metastasis	SiHa, CaSki, human patient and mice tissues/cells	[248]
			↑FAK and ↑Pyk2 ->↓focal adhesions ->	migration, metastasis		
			S and G2/M phases (↓p21, ↑Cdc25C) ->	proliferation		
			VEGF (vascular endothelial growth factor) ->	angiogenesis		
**Colorectal**		↑STIM1	↑EGF -> ↑COX-2 -> ↑PGE2 ->	migration, EMT	DDL-1, HT-29, patient samples	[85]
		↑Orai1, Orai3	hypoxia -> ↑HIF-1/2a -> ↑Orai3 ->	migration	CW-2	[275]
		↓STIM2 + ↑TRPC1 + ↑Orai1 + ↑STIM1	n.d.	proliferation, invasion, survival, apoptosis	HT29	[254]
**Esophageal**		↑Orai1 (STIM2?)	↑ERK & Akt -> cell cycle (↓cdc2, ↓cyclin B1, ↓p27) ->	proliferation	KYSE-150, patient and mouse samples	[255]
			↑vimentin, ↑Rac1, ↓E-cadherin -> cytoskeleton ->	migration, invasion		
**Gastric**		↑STIM1, ↑Orai1	cell cycle (↑cyclin D1, ↓p21) ->	proliferation, metabolism, migration, invasion, metastasis	GS, BGC-803, BGC-823, MGC-803, MKN-28, MKN-45, SGC-7901, nude mice, patient samples	[53]
			↑vimentin, ↑fibronectin, ↑MACC1, ↓E-cadherin ->	migration, metastasis		
**Glioblastoma**		↑STIM1	cell cycle (G0/G1 phase, ↑cyclin D1, ↑CDK4, ↓p21) ->	proliferation	U251, U87 and U373	[88]
		↑Orai1	↑Pyk2 -> ↓focal adhesion -> ↑vimentin, ↓E-cadherin, ↑N-cadherin (EMT like) ->	migration, invasion	U251/SNB19	[272]
**Hematologic**	AML	↑Orai1, ↑Orai2	↑FAK -> ↓focal adhesions ->	proliferation, migration	HL60	[245]
		Orai3	↑Ras -> ↑Orai3 ->	cell survival	U937, 8226	[262]
	MM	↑STIM1, ↑Orai1	↑cell cycle ->	proliferation, apoptosis	KM3, U266	[261]
	CLL	↑STIM1, ↑Orai1, ↑TRPC1	n.d.	proliferation, cancer progression	U937, 8226	[89]
**Liver**		↑STIM1	↑FAK-Y397 -> ↓focal adhesions ->	migration	HCC-LM3	[257]
		↑STIM1, ↑Orai1, ↑TRPC6	cell cycle (↑cyclin D1) ->	proliferation	Huh-7	[258]
**Lung**		↓Orai1	↑EGF -> ↑PI3K/Akt -> cell cycle (G1/S phase; ↑cyclin D) ->	proliferation	A549	[79]
		↑Orai3/Orai1	↑EGF -> PI3K/Akt -> cell cycle (G1/S phase; ↑cyclin D1/E, ↑CDK4 and ↑CDK2) ->	proliferation, cell cycle progression	NCI-H23, NCI-H460, patients	[249]
**Melanoma**		↑STIM1, ↑Orai1	↑CaMKII/Raf-1/ERK ->	proliferation, migration, metastasis	SK-Mel-2, C8161, SK-Mel-24, UACC2577, WM3248,	[276]
		↑STIM2, ↑Orai1	n.d.	migration, invasion	SK-MEL-5, SK-MEL-28, WM3734	[108]
		↓STIM2, ↓Orai1	CREB/β-catenin -> MITF	proliferation		
**Ovarian**		↑STIM1, ↑Orai1	↑Akt ->	apoptosis	A2780	[264]
		↑TRPC1, ↑TRPC3, ↑TRPC4, ↑TRPC6	RTK? ->	proliferation	SKOV3, ATCC HTB-77	[263]
**Pancreatic**		↑STIM1, ↑Orai1	n.d.	apoptosis	Panc1, (ASPC1, BxPC3, MiaPaca2, Capan1)	[265]
		↑Orai3	↑cell cycle (G2/M-phase)	proliferation	Panc1, (ASPC1, BxPC3, MiaPaCa2, Capan1)	[266]
**Prostate**		↓Orai1	↓AR	apoptosis	LNCaP, DU-145, and PC-3	[80]
		↑Orai1/Orai3	cell cycle (G1/S phase; ↑cyclin D)	proliferation	LNCaP	[269]
			↓SOCE ->	apoptosis		
**Renal**		↑STIM1, ↑Orai1	n.d.	proliferation, migration	ccRCC, ACHN and Caki1, patient samples	[277]

Symbol: ? … unknown role or pathway.

**Table 2 cancers-15-00101-t002:** Correlation of cancer types with critical SK channels.

Cancer Type	Critical Proteins	Targeted Signaling Pathways	Affected Cancer Hallmarks	Cell Type	Ref
**breast**	↑SK3	n.d.	migration, metastasis	MDA-MB-435s	[327]
	↑SK4	cell cycle (G1, S phases), ↑cdc25C	proliferation	mice	[360]
		EGF -> vimentin, snail1	proliferation, migration, EMT	MCF-7, T47D, MDA-MB-231 and MDA-MB-468, patients	[358]
	↑SK3, ↑P2X7R	hypoxia ->↑ERK1/2 ->↑Akt (?)	proliferation, invasiveness, migration	MDA-MB-435s	[330]
**colorectal**	↑SK4	↑Ras/ERK (KRAS)/HIF1a/ROS	migration, invasion, metastasis	HCT116	[359]
**glioblastoma**	↑SK4	↑cell cycle (G2, M phases)	invasion, proliferation, poor prognosis	T98G, U87MG, GL261, patients	[362]
**hematologic CCL**	↑SK4	n.d.	proliferation	patients	[352]
**melanoma**	↑SK3	n.d.	migration, metastasis	518A2, HBL, Bris	[342]
**ovarian**	↑SK4, P2y_2_	n.d.	proliferation, migration, cancer progression	Skov-3, patients	[361,363]
**pancreatic**	↑SK4	↑KRAS -> ↑RAS -> ↑ERK/PI3K	proliferation, poor prognosis	mice, patients	[356]
**prostate**	↑SK3 (↑ZEB1)	↑Snail, ↑Slug, ↑Twist	EMT, neuroendocrine differentiation, drug resistance	LNCaP, patients	[331,364]

Symbol: ? … unknown role or pathway.

**Table 3 cancers-15-00101-t003:** Correlation of cancer types with detected SK3–Orai1 interplay and accessory proteins, targeted signaling pathways, types of affected cancer hallmarks, and corresponding cell type.

Cancer Type	Critical Proteins	Targeted Signaling Pathways	Affected Cancer Hallmarks	Cell Type	Ref
**Breast**	SK3 + ↑Orai1	cAMP-PKA (↓SK3 activity due to phosphorylation)	migration, metastasis	MDA-MB-435s	[322]
	SK3 + ↑Orai1 + ↑SigmaR1	n.d.	migration	MDA-MB-435	[321]
**colon**	SK3 + ↑Orai1 + ↑SigmaR1	n.d.	migration	HCT-116, patients	[321]
	↑SK3 + ↑Orai1 + ↑TRPC1 + ↑STIM1	↑EGFR -> ↑PI3K -> ↑Akt -> ↑Rac-1 -> ↑Calpain	migration	HCT-116	[323]
		↑EGFR -> ↑PI3K -> ↑Akt -> STIM1	migration		
**prostate**	SK3 + Orai1	n.d.	proliferation	LNCaP	[343]

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
