# Peer review of "CRAC and SK Channels: Their Molecular Mechanisms Associated with Cancer Cell Development"

_cancers, 2022, doi:10.3390/cancers15010101_

Round 1

Reviewer 1 Report

The review by Tiffner et al is an extensive and timely one on the roles of CRAC and SK channels in the development of cancer cells. The models and tables shown are informative. Nonetheless, the manuscript would benefit from improvements on the following points:

1.    It can be a rather laborious read in some parts because of the grammar and how the sentence/paragraph is structures. The manuscript would benefit from more cohesiveness in this area as the different writing styles of each author are noticeable in parts.

2.    There are "paragraphs" containing fewer than 5 sentences. The author should consider incorporating these into the preceding or following paragraph. Alternatively, if there is a series of such "paragraphs" in a section, the authors should consider combining them to form one paragraph .

3.    Title of the review is “CRAC & SK channels: Their molecular mechanisms associated 2 with cancer cell development”.

Doesn’t CRAC generally refers to STIM1/Orai1 combo?  The review includes STIM2 and also Orais 2 and 3.

4.    Abstract:

a.    Reading of its present form is rather stilted. Can the authors work on the flow?

b.    Lines 21 to 23:  This part is not clear and "coregulation" seems a bit of a misnomer when referring to Ca2+. Do the authors mean the relationship between plasma membrane channels that mediate extracellular Ca2+ influx and those dependent on Ca2+ for their activity? If the word "Ca2+" is more general and refers to channels and pumps that mediate influx, sequestration and efflux (e.g. via pumps and/or exchangers), please re-write the sentence to improve clarity.

5.    Figure 1:

a.    What to the lines in the bottom image of Fig. 1C represent? They look like fibers protruding out from the cell. Are these lamellopodia? Would be good to label them on the image.

b.    Figure legend is really long. If the models are referred to and further described in the main text, the figure legend can be shortened to provide only a description of what each model shows.

c.    Line 120: Please put the abbreviation MET in brackets next to the word “metastasis”

6.    Line 156: “deregulate” refers to the removal of regulations or restrictions. I am not sure how this can happen in a bidirectional manner. Do the authors mean that Ca2+ can either enhance or downregulate the Ras-ERK and Akt pathways? Please revise the sentence.

7.    Line 159: “Vice versa”  is not used to start a sentence. The authors should use “Conversely” or another synonym.

8.    Line 161: “Additionally, Ca2+ affects the PI3K/Akt pathway.”

How? An explanatory sentence would be better used to show how Ca2+ positively or negatively impacts the pathway.

Also, is this a repetition of what is already stated in Line 156? If yes, please revise to minimize repetitions.

9.    Line 232: Instead of inset, use “Top”, as in the figure legend.

10.  Line 262: see point #7

11.  Figure 3:

a.    (A) – small text on the size is difficult to read. Please enlarge if possible.

b.    (A) – the color for the line showing the C-terminal region after the CC domains are different in the Resting state vs Activated state (looks darker, difficult to see). Please use the same light color (as shown in the Resting state).

c.    (B) – small text on the size is difficult to read. Please enlarge if possible.

12.  Line 358: “Especially,” does not seem right since STIM1 and STIM2 are not the only proteins ubiquitously expressed in many tissues. Please revise the sentence.

13.  Line 403: Missing a comma – “In the tightly packed, inactive conformation, ...”

14.  Line 412: "Exemplarily," means commendable. I think the authors meant "For example,".

15.  Lines 482-483: “Also in cancer, the remodelling of Ca2+ ion channels may in-482 volve their altered arrangement and interplay in the membrane (3)”

Can the authors clarify what they meant by altered arrangement and interplay? Are the authors referring to: (1) structural rearrangments of the channel; (2) assembly of the channel complex; (3) channel interaction with regulatory proteins that affect their function; (4) other?

16.  Line 493: “…in the PM, which is enriched in cholesterol-rich regions (181,191,194–197).”

17.  Lines 511 to 512: I think the authors meant to say endogenous SOCE was reduced by MβCD treatment but enhanced by cholesterol oxidase or fillipin. Is this correct? Please rectify sentence as needed.

18.  Lines 518 to 519: Perturbations to plasma membrane lipids also affect other channels and/or proteins that impact STIM1 and Orai1 expression and function. The authors should briefly mention this possibility in their review, and refer readers to other papers for more details as needed.

19.  Titles for Tables 1, 2 and 3: Maybe shorten the title  since the following sentence basically explains what the table contains. For example, Table 1: "Correlation of different cancer types with critical CRAC channel components."

20.  Figure 5: Small text in the images make it difficult to read. Please enlarge if possible, or increase the resolution to make it sharper.

21.  Lines 984 to 985: "forced" sounds awkward; maybe use "overexpression"?

22.  Lines 1010 to 1011: “However, peptides arose as limited in 1010 medical use due to their high toxicity or low efficiency.

I am not sure which group has limited medical use - peptides, synthetic analogues or modulators. Please revise this sentence.

23.  Line 1063: Should be “SK3-Orai interplay” – missing a hyphen

24.  Line 1121: Instead of “Supportively,..”, consider using "Additionally" or "Moreover”.

25.  Lines 1150 to 1153: If STIM1 coupling with Orai1 is not required for Orai1-SK3 interplay, can Orai1 function in a store-independent manner and still interact with SK3? Doesn’t CRAC generally refers to STIM1/Orai1 complex in a store-dependent manner?

26.  Lines 1167 to 1169: Please cite the paper for this observation at the end of the sentence.

Author Response

Our reply is attached.

Reviewer 2 Report

This is a well-written review article. The only minor suggestion to the authors is to consider including a brief description of recently reported observations on the role of protein kinase CK2 in Ca2+ signaling in prostate cancer cells (reported in Afzal M, et al. (2020) Protein kinase CK2 impact on intracellular calcium homeostasis in prostate cancer. Mol Cell Biochem 470:131–143. https:// doi. org/ 10. 1007/ s11010- 020- 03752-4).

.

Author Response

Our reply is attached.

Reviewer 3 Report

This manuscript was designed to review the role of CRAC and SK channels in cancer development. CRAC and SK channels play an important role in regulating calcium signaling in normal and cancer cells. The review summarizes our current understanding of the topic and it is clearly written. Below is a list of issues that I considered should be addressed before the manuscript is published in Cancers.

1) Page 7, line 224: I wonder if the authors can expand on the role of Ca2+ ions in regulating EMT. For example, how do Ca2+ ions regulate changes in E-/N-cadherin expression (or any other associated protein involved in EMT).

2) Page 7, line 255: The authors should expand on the role of hypoxia in regulating cancer development. For example, can they describe specific cancer examples where this regulation has been established?

3)     Page 9, Fig.3A: for readers not familiar with the topic, please label the STIM and ORAI proteins in Figure 3A.

4)     Page 25, line 928: the references about the expression of SK channels in prostate cancer cells are not appropriated. The only references that establishes a potential role of SK channels in prostate cancer cells are Parihar et al., Eur J Pharmacol. (2003) and Bloch et al. Oncogene (2007).

5)     There are many grammatical errors throughout the text:

a)     Page 2, line 36, missing comma: genetic and epigenetic alterations, such as dysregulation of SK3 and Orai1 channels…

a)     Page 2, line 41, missing comma: epithelial-mesenchymal transition of the cell, promoting migration…

b)     Page 3, line 72, substitute to by that: Ca2+ ions are versatile intracellular signals that regulate a plethora of cellular processes.

c)     Page 3, line 62, missing comma: One essential factor, which strongly contributes to cancer progression…

d)     Page 3, lines 83-86, should be part of the paragraph above.

e)     Page 3, line 88, missing comma: cellular factors (e.g. inositol-tri-phosphate (IP3), diacylglycerol (DAG)), which activate…

f)      Page 5, line 111, substitute at by in: In the mitochondria, …

g)     Page 5, line 154, missing comma: receptor stimulation triggers the activation of the lipid kinase (PI3K), which in consequence activates…

h)     Page 10, line 389, missing comma: an extended conformation, which is stabilized by…

i)      Page 17, line 583, missing comma: the canonical transient receptor potential channels, which have…

j)      Page 18, line 637, missing comma: intracellular Ca2+ concentration, which controls cell proliferation…

k)     Page 27, line 1027, missing comma: lipid membrane reorganization, which inhibits SK3 channels…

l)      That authors should consider situations where the pronoun that should be used instead of which.

Author Response

Our reply is attached.
